# UNI-MOL: A UNIVERSAL 3D MOLECULAR REPRESENTATION LEARNING FRAMEWORK

**Gengmo Zhou**[1,2]* **Zhifeng Gao**[2]*† **Qiankun Ding**[2] **Hang Zheng**[2]
**Hongteng Xu**[1] **Zhewei Wei**[1] **Linfeng Zhang**[2,3] **Guolin Ke**[2]†
[1]Renmin University of China    [2]DP Technology    [3]AI for Science Institute, Beijing
{zgm2015, hongtengxu, zhewei}@ruc.edu.cn
{gaozf, dingqk, zhengh, zhanglf, kegl}@dp.tech

## ABSTRACT

Molecular representation learning (MRL) has gained tremendous attention due to its critical role in learning from limited supervised data for applications like drug design. In most MRL methods, molecules are treated as 1D sequential tokens or 2D topology graphs, limiting their ability to incorporate 3D information for downstream tasks and, in particular, making it almost impossible for 3D geometry prediction/generation. In this paper, we propose a universal 3D MRL framework, called Uni-Mol, that significantly enlarges the representation ability and application scope of MRL schemes. Uni-Mol contains two pretrained models with the same SE(3) Transformer architecture: a molecular model pretrained by 209M molecular conformations; a pocket model pretrained by 3M candidate protein pocket data. Besides, Uni-Mol contains several finetuning strategies to apply the pretrained models to various downstream tasks. By properly incorporating 3D information, Uni-Mol outperforms SOTA in 14/15 molecular property prediction tasks. Moreover, Uni-Mol achieves superior performance in 3D spatial tasks, including protein-ligand binding pose prediction, molecular conformation generation, etc. The code, model, and data are made publicly available at `https://github.com/dptech-corp/Uni-Mol`.

## 1 INTRODUCTION

Recently, representation learning (or pretraining, self-supervised learning) [1; 2; 3] has been prevailing in many applications, such as BERT [4] and GPT [5; 6; 7] in Natural Language Processing (NLP), ViT [8] in Computer Vision (CV), etc. These applications have a common characteristic: unlabeled data is abundant, while labeled data is limited. As a solution, in a typical representation learning method, one first adopts a pretraining procedure to learn a good representation from large-scale unlabeled data. Then a finetuning scheme is followed to extract more information from limited supervised data.

Applications in the field of drug design share the characteristic that calls for representation learning schemes. The chemical space that a drug candidate lies in is vast, while drug-related labeled data is limited. Not surprisingly, compared with traditional molecular fingerprint-based models [9; 10], recent molecular representation learning (MRL) models perform much better in most property prediction tasks [11; 12; 13]. However, to further improve the performance and extend the application scope of existing MRL models, one is faced with a critical issue. From the perspective of life science, the properties of molecules and the effects of drugs are mostly determined by their 3D structures [14; 15]. In most current MRL methods, one starts with representing molecules as 1D sequential strings, such as SMILES [16; 17; 18] and InChI [19; 20; 21], or 2D graphs [22; 11; 23; 12; 24]. This may limit their ability to incorporate 3D information for downstream tasks. In particular, this makes it almost impossible for 3D geometry prediction or generation, such as, e.g., the prediction of protein-ligand binding pose [25]. Even though there have been some recent attempts trying to leverage 3D information in MRL [26; 27], the performance is less than optimal, possibly due to the small size of

---

*Equal contribution.
†Corresponding authors.

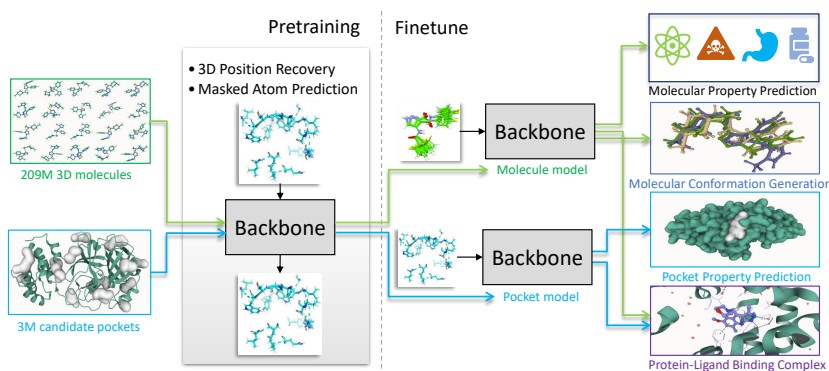

Figure 1: Schematic illustration of the Uni-Mol framework.

3D datasets, and 3D positions can not be used as inputs/outputs during finetuning, since they only serve as auxiliary information.

In this work, we propose Uni-Mol, to our best knowledge, the first universal 3D molecular pretraining framework, which is derived from large-scale unlabeled data and is able to directly take 3D positions as both inputs and outputs. In particular, Uni-Mol consists of 3 parts. 1) *Backbone*. A Transformer based model that can effectively capture the input 3D information, and predict 3D positions directly. 2) *Pretraining*. Two large-scale datasets: a 209M molecular conformation dataset and a 3M candidate protein pocket dataset, for pretraining 2 models on molecules and protein pockets, respectively. And two pretraining tasks: 3D position recovery and masked atom prediction, for effectively learning 3D spatial representation. 3) *Finetuning*. Several finetuning strategies for various downstream tasks. For example, how to use the pretrained molecular model in molecular property prediction tasks; how to combine the two pretrained models in protein-ligand binding pose prediction. We refer to Fig. 1 for an overall schematic illustration of the Uni-Mol framework, and the details will be described in Sec. 2.

To demonstrate the effectiveness of Uni-Mol, we conduct experiments on a series of downstream tasks. In the molecular property prediction tasks, Uni-Mol outperforms SOTA on 14/15 datasets on the MoleculeNet benchmark. In 3D geometric tasks, Uni-Mol also achieves superior performance. For the pose prediction of protein-ligand complexes, Uni-Mol predicts 80.35% binding poses with RMSD <= 2Å, 22.58% relatively better than popular docking methods, and ranks 1st in the docking power test on CASF-2016 [28] benchmark. Regarding molecular conformation generation, Uni-Mol achieves SOTA for both Coverage and Matching metrics on GEOM-QM9 and GEOM-Drugs [29]. Moreover, Uni-Mol can be successfully applied to tasks with very limited data like pocket druggability prediction.

To summarize, Uni-Mol made the following contributions: 1) To our best knowledge, Uni-Mol is the first pure 3D molecular pretraining framework that can predict 3D positions, and the first molecular pretraining framework that can be directly used in 3D tasks in the field of drug design. 2) Based on extensive benchmarks, we build a simple and efficient SE(3) Transformer backbone[1], and an effective 3D pretraining strategy in Uni-Mol. 3) Uni-Mol outperforms SOTA in various downstream tasks. 4) The whole Uni-Mol framework, including code, model, and data, will be made publicly available.

## 2 UNI-MOL FRAMEWORK

### 2.1 BACKBONE

In MRL, there are two well-known backbone models, graph neural networks(GNN) [22; 23; 12] and Transformer [24; 11]. With GNN as the backbone model, for efficiency purposes, locally connected graphs are often used to represent molecules. However, the locally connected graph lacks the ability to capture the long-range interactions among atoms. We believe that long-range interactions are important in MRL. Therefore, We choose Transformer as the backbone model in Uni-Mol, as it fully connects the nodes/atoms and thus can learn the possible long-range interactions.

---

[1]Although the backbone can output SE(3)-equivariant positions, it is based on SE(3)-invariant representations.

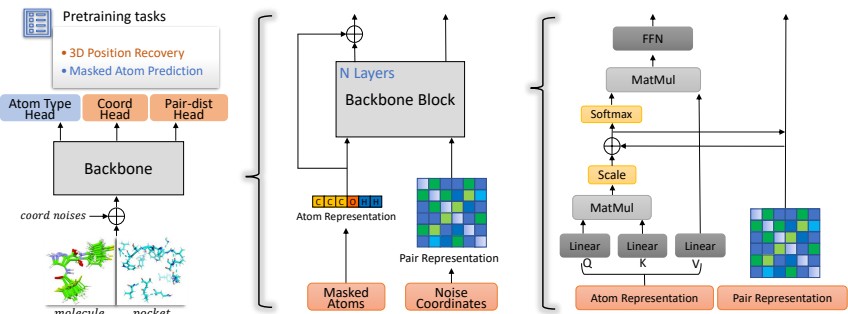

Figure 2: Left: the overall pretraining architecture. Middle: the model inputs, including atom representation and pair representation. Right: details in the model block.

However, Transformer cannot handle 3D spatial data directly as it was originally designed for NLP tasks. Although there are several recent works extending Transformer to 3D data [30; 31], most of them are much slower than the standard Transformer due to the complex additional components like Tensor Field Networks. Considering the pretraining cost in the large-scale dataset, we need an efficient backbone. To achieve that, based on the standard Transformer with Pre-LayerNorm [32], we introduce several efficient and necessary modifications, for the ability to take 3D positions as inputs and outputs.

**Architecture Overview** As illustrated in Fig. 2, the Uni-Mol backbone is a Transformer based model. It has two inputs, atom types and atom coordinates. And two representations (atom and pair) are maintained in the model. The atom representation is initialized from atom types, by the Embedding layer; The pair representation is initialized by invariant spatial positional encoding calculated from atom coordinates. In particular, based on pair-wise Euclidean distances among atoms, the pair representation is invariant to global rotation and translation. The two representations communicate with each other in self-attention module. Details are in the following subsections.

**Encode 3D positions** Due to its permutational invariance, Transformer cannot distinguish the positions of inputs without positional encoding. Different with the discrete positions used in NLP/CV [33; 34], the positions in 3D space, i.e. coordinates, are continuous values. Besides, the positional encoding procedure needs to be invariant under global rotation and translation. Several 3D spatial positional encodings were already proposed to tackle this [35; 36; 37; 38], and we have no interest in reinventing a new one. Therefore, we benchmark existing encodings (in Appendix D.1), and use a simple and effective one: Euclidean distances of atom pairs, followed by a pair-type aware Gaussian kernel [39].

Furthermore, since the invariant 3D spatial positional encoding is encoded at the pair level, we also maintain a pair-level representation in Transformer, to enhance the 3D spatial representation. Specifically, the pair representation is initialized as the aforementioned spatial positional encoding. Then, to update pair representation, we use atom-to-pair communication via the result of the multi-head Query-Key product in self-attention. Formally, the update of $ij$ pair representation is denoted as

$$\boldsymbol{q}_{ij}^{l+1} = \boldsymbol{q}_{ij}^{l} + \{\frac{\boldsymbol{Q}_i^{l,h}(\boldsymbol{K}_j^{l,h})^T}{\sqrt{d}}|h \in [1, H]\}, \tag{1}$$

where $\boldsymbol{q}_{ij}^{l}$ is the pair representation of atom pair $ij$ in $l$-th layer, $H$ is the number of attention heads, $d$ is the dimension of hidden representations, and $\boldsymbol{Q}_i^{l,h}$ ($\boldsymbol{K}_j^{l,h}$) is the Query (Key) of the $i$-th ($j$-th) atom in the $l$-th layer $h$-th head. Besides, to leverage 3D information in the atom representation, we also introduce pair-to-atom communication, by using the pair representation as the bias term in self-attention. Formally, the self-attention with pair-to-atom communication is denoted as

$$\text{Attention}(\boldsymbol{Q}_i^{l,h}, \boldsymbol{K}_j^{l,h}, \boldsymbol{V}_j^{l,h}) = \text{softmax}(\frac{\boldsymbol{Q}_i^{l,h}(\boldsymbol{K}_j^{l,h})^T}{\sqrt{d}} + \boldsymbol{q}_{ij}^{l-1,h})\boldsymbol{V}_j^{l,h}, \tag{2}$$

where $\boldsymbol{V}_j^{l,h}$ is the Value of the $j$-th atom in the $l$-th layer $h$-th head. As shown in the above equations, the proposed pair representation is very simple, and the extra cost of maintaining it is negligible. And our benchmark in Appendix D also demonstrates its efficiency and effectiveness.

---

**Algorithm 1** Corrupted Position Generation and Assignment.

---

**Require:** $\mathbf{X} \in \mathbb{R}^{m \times 3}$: coordinates of $m$ atoms, $r$: noise range, re_assign: use re-assignment or not
1: $\mathbf{R} = \mathbf{X} + \delta$, where $\delta \sim \text{Uniform}(-r\text{Å}, r\text{Å})$          ▷ Generate corrupted positions
2: **if** (not re_assign) **return** $\mathbf{R}$          ▷ Return directly if not need the re-assignment
3: $\mathbf{D} = \{D[i,\ j] = \|X_i - R_j\|_2 \mid 1 \le i, j \le m\}$          ▷ Compute distance
4: **for** $i$ in **random_perm**$(1, m)$ **do**          ▷ Greedy assignment based a random order
5:      $k = \text{argmin}(\mathbf{D}_{i,:})$          ▷ Get the nearest position at the $i$-th row
6:      $\mathbf{Y}_i = \mathbf{R}_k$          ▷ Assignment
7:      $\mathbf{D}_{:,k} = \inf$          ▷ Marked the $k$-th column as used
    **return** $\mathbf{Y}$          ▷ Return corrupted positions

---

**Predict 3D positions** With 3D spatial positional encoding and pair representation, the model can learn a good 3D representation. However, it still lacks the ability to directly output coordinates, which is essential in 3D spatial tasks. To this end, we introduce an SE(3)-equivariant head to predict the delta positions based on SE(3)-invariant pair representation and equivariant input $\boldsymbol{x}_i - \boldsymbol{x}_j$, denoted as

$$\hat{\boldsymbol{x}}_i = \boldsymbol{x}_i + \sum_{j=1}^{n} \frac{(\boldsymbol{x}_i - \boldsymbol{x}_j)c_{ij}}{n}, \quad c_{ij} = \text{ReLU}((\boldsymbol{q}_{ij}^L - \boldsymbol{q}_{ij}^0)\boldsymbol{U})\boldsymbol{W}, \tag{3}$$

where $n$ is the number of atoms, $L$ is the number of layers , $\boldsymbol{x}_i \in \mathbb{R}^3$ is the input coordinate of $i$-th atom, and $\hat{\boldsymbol{x}}_i \in \mathbb{R}^3$ is the output coordinate of $i$-th atom, $\text{ReLU}(y) = \max(0, y)$ is Rectified Linear Unit [40], $\boldsymbol{U} \in \mathbb{R}^{H \times H}$ and $\boldsymbol{W} \in \mathbb{R}^{H \times 1}$ are the projection matrices to convert pair representation to scalar. This head is similar to the position update procedure in EGNN [41], but much more efficient due to Uni-Mol only updating 3D positions in the last layer. Besides, to be consistent with delta position prediction, Uni-Mol uses delta pair representation to update coordinates, while EGNN uses pair representation directly. Our benchmark in Appendix D.3 demonstrates the one in Uni-Mol is better.

Please note that the backbone in Uni-Mol can be replaced with any SE(3) model that can take 3D positions as inputs and outputs. However, considering the massive pretraining cost in the large-scale dataset, we favor the efficient backbone. So we only make several simple and necessary modifications to the standard Transformer model, based on the ablation benchmark results in Appendix D.

## 2.2 PRETRAINING

**Large-Scale dataset** For the purpose of pretraining, we generate two large-scale datasets, one composed of 3D structures of organic molecules, and another composed of 3D structures of candidate protein pockets. Then, two models are pretrained using these two datasets, respectively. As pockets are directly involved in many drug design tasks, intuitively, the pretraining on candidate protein pockets can boost the performance of tasks related to protein-ligand structures and interactions.

The molecular pretraining dataset is based on multiple public datasets (See Appendix A for more information). After normalizing and deduplicating, it contains about 19M molecules. To generate 3D conformations, we use ETKGD [42] with Merck Molecular Force Field [43] optimization in RDKit [44] to randomly generate 11 conformations for each molecule, totally 209M conformations.

The protein pocket pretraining dataset is derived from the Protein Data Bank (RCSB PDB) [45], a collection of 180K 3D structures of proteins. To extract candidate pockets, we first clean the data by adding the missing side chains and hydrogen atoms; then we use Fpocket [46] to detect possible binding pockets of the proteins; and finally, we filter pockets by the number of residues. In this way, We collect a dataset of 3.2M candidate pockets for pretraining.

**Pretraining strategies** Self-supervised task is vitally important for effective learning from large-scale unlabeled data. For example, the masked token prediction task in BERT [4] encourages the model to learn the contextual information. In Uni-Mol, we want to encourage the model to learn the 3D structural information during pretraining. To this end, we design a 3D position recovery self-supervised task. The main idea of the task is to recover the correct 3D positions, given the corrupted input positions. An intuitive way is to mask the positions, like the token masking in BERT.

However, the positions are continuous values, not discrete values; we cannot use a special value to represent the mask (like the `[MASK]` token in BERT).

Therefore, rather than masking, random positions are used as corrupted input 3D positions, and the model is trained to predict the correct position. Nevertheless, learning the mapping from a random position to the ground-truth atom position is very challenging. There are two technologies to reduce the delta positions (between random and ground-truth positions), making the learning more feasible. First, *re-assignment*, given $m$ atoms and $m$ random positions, there are $m!$ possible assignments. Among them, following the stationary-action principle [47], we can use the one with *minimal* delta positions. Due to the difficulty of finding an optimal solution, we use an efficient greedy algorithm to find a sub-optimal re-assignment. Second, *noise range*, we can limit the space of the random positions, only allowing the random positions with a noise ($r$) around the ground-truth ones. There is a tradeoff here; if $r$ is large, the re-assignment is required to make the learning feasible; if $r$ is small, the re-assignment may not need. We summarized the algorithm into Alg. 1, and benchmarked several settings, details in Appendix D.6, and found a simple and effective one: use $r = 1$ Å, without re-assignment.

Then, with corrupted input coordinates, two additional heads are used to recover the correct positions. 1) Pair-distance prediction. Based on pair-representation, the model needs to predict the correct Euclidean distances of the corrupted atoms pairs. 2) Coordinate prediction. Based on SE(3)-Equivariant coordinate head, the model needs to predict the correct coordinates for the corrupted atoms.

Finally, the atom types for the corrupted atoms are masked, and a head is used to predict the correct atom types. For the convenience of finetuning, similar to BERT, a special atom `[CLS]`, whose coordinate is the center of all atoms, is used to represent the whole molecule/pocket. Both 2 pretraining models use the same self-supervised tasks described above, and Figure 2 is an illustration of the overall pretraining framework. For the detailed configurations of pretraining, please refer to Appendix C. Similar to backbone design, we conduct an extensive benchmark for pretraining strategies (in Appendix D.5 and D.6), and choose the above strategies based on performance.

## 2.3 FINETUNING

To be consistent with pretraining, we use the same data prepossessing pipeline during finetuning. For molecules, as multiple random conformations can be generated in a short time, we can use them as data augmentation in finetuning to improve performance and robustness. For tasks that provide atom coordinates, we use them directly and skip the 3D conformation generation process. As there are 2 pretraining models and several types of downstream tasks, we should properly use them in the finetuning stage. According to the task types, and the involvement of protein or ligand, we can categorize them as follow.

**Non-3D prediction tasks** These tasks do not need to output 3D conformations. Examples include molecular property prediction, molecule similarity, pocket druggability prediction, protein-ligand binding affinity prediction, etc. Similar to NLP/CV, we can simply use the representation of `[CLS]`, which represents the whole molecule/pocket, or the mean representation of all atoms, with a linear head to finetune on downstream tasks. In the tasks with pocket-molecule pair, we can concatenate their `[CLS]` representations, and then finetune with linear head.

**3D prediction tasks of molecules or pockets** These tasks need to predict a 3D conformation of the input, such as molecular conformation generation. Different from the fast conformation generation method used in Uni-Mol, molecular conformation generation task usually requires running advanced sampling and semi-empirical density functional theory (DFT) to account for the ensemble of 3D conformers that are accessible to a molecule, and this is very time-consuming. Therefore, there are many recent works that train the model to fast generate conformations from molecular graph [48; 49; 50; 51]. While in Uni-Mol, this task straightforwardly becomes a conformation optimization task: generate a new conformation based on a different input conformation. Specifically, in finetuning, the model supervised learns the mapping from Uni-Mol generated conformations to the labeled conformations. Moreover, the output conformations can be generated end-to-end by SE(3)-Equivariant head.

**3D prediction tasks of protein-ligand pairs** This is one of the most important tasks in structure-based drug design. The task is to predict the complex structure of a protein binding site and a

Table 1: Uni-Mol performance on molecular property prediction classification tasks

| | Classification (ROC-AUC %, higher is better ↑) | | | | | | | | |
|---|---|---|---|---|---|---|---|---|---|
| Datasets
# Molecules
# Tasks | BBBP
2039
1 | BACE
1513
1 | ClinTox
1478
2 | Tox21
7831
12 | ToxCast
8575
617 | SIDER
1427
27 | HIV
41127
1 | PCBA
437929
128 | MUV
93087
17 |
| D-MPNN | 71.0(0.3) | 80.9(0.6) | 90.6(0.6) | 75.9(0.7) | 65.5(0.3) | 57.0(0.7) | 77.1(0.5) | 86.2(0.1) | 78.6(1.4) |
| Attentive FP | 64.3(1.8) | 78.4(0.022) | 84.7(0.3) | 76.1(0.5) | 63.7(0.2) | 60.6(3.2) | 75.7(1.4) | 80.1(1.4) | 76.6(1.5) |
| N-Gram$_{RF}$ | 69.7(0.6) | 77.9(1.5) | 77.5(4.0) | 74.3(0.4) | - | 66.8(0.7) | 77.2(0.1) | - | 76.9(0.7) |
| N-Gram$_{XGB}$ | 69.1(0.8) | 79.1(1.3) | 87.5(2.7) | 75.8(0.9) | - | 65.5(0.7) | 78.7(0.4) | - | 74.8(0.2) |
| PretrainGNN | 68.7(1.3) | 84.5(0.7) | 72.6(1.5) | 78.1(0.6) | 65.7(0.6) | 62.7(0.8) | 79.9(0.7) | 86.0(0.1) | 81.3(2.1) |
| GROVER$_{base}$ | 70.0(0.1) | 82.6(0.7) | 81.2(3.0) | 74.3(0.1) | 65.4(0.4) | 64.8(0.6) | 62.5(0.9) | 76.5(2.1) | 67.3(1.8) |
| GROVER$_{large}$ | 69.5(0.1) | 81.0(1.4) | 76.2(3.7) | 73.5(0.1) | 65.3(0.5) | 65.4(0.1) | 68.2(1.1) | 83.0(0.4) | 67.3(1.8) |
| GraphMVP | 72.4(1.6) | 81.2(0.9) | 79.1(2.8) | 75.9(0.5) | 63.1(0.4) | 63.9(1.2) | 77.0(1.2) | - | 77.7(0.6) |
| MolCLR | 72.2(2.1) | 82.4(0.9) | 91.2(3.5) | 75.0(0.2) | - | 58.9(1.4) | 78.1(0.5) | - | 79.6(1.9) |
| GEM | 72.4(0.4) | 85.6(1.1) | 90.1(1.3) | 78.1(0.1) | 69.2(0.4) | **67.2(0.4)** | 80.6(0.9) | 86.6(0.1) | 81.7(0.5) |
| Uni-Mol | **72.9(0.6)** | **85.7(0.2)** | **91.9(1.8)** | **79.6(0.5)** | **69.6(0.1)** | 65.9(1.3) | **80.8(0.3)** | **88.5(0.1)** | **82.1(1.3)** |

molecular ligand. Besides the conformation changes of the pocket and the molecule themselves, we also need to consider how the molecule lays in the pocket, that is, the additional 6 degrees (3 rotations and 3 translations) of freedom of a rigid movement. In principle, with Uni-Mol, we can predict the complex conformation by the SE(3)-Equivariant head in an end-to-end fashion. However, this is unstable as it is very sensitive to the initial docking positions of molecular ligands. Herein, to get rid of the initial positions, we use a scoring function based optimization method in this paper. In particular, the molecular representation and pocket representation are firstly obtained from their own pretraining models by their own conformations; then, their representations are concatenated as the input of an additional 4-layer Uni-Mol decoder, which is finetuned to learn the pair distances of all heavy atoms in molecule and pocket. Then, with the predicted pair-distance matrix as a scoring function, we first randomly place the ligand and then optimize the coordinates of its atoms by directly back-propagation the loss of the current pair-distance matrix and the predicted pair-distance matrix. Thanks to the efficiency of back-propagation, this process is very fast, about 100x faster than traditional docking tools. More details can be found in Appendix C.6.

## 3 EXPERIMENTS

To verify the effectiveness of our proposed Uni-Mol model, we conduct extensive experiments on multiple downstream tasks, including molecular property prediction, molecular conformation generation, pocket property prediction, and protein-ligand binding pose prediction. Besides, we also conduct several ablation studies. Due to space restrictions, we leave the detailed experimental settings and ablation studies to Appendix C and D.

### 3.1 MOLECULAR PROPERTY PREDICTION

**Datasets and setup**  MoleculeNet [52] is a popular benchmark for molecular property prediction, including datasets focusing on different molecular properties, from quantum mechanics and physical chemistry to biophysics and physiology. Following previous work GEM [13], we use scaffold splitting and report the mean and standard deviation by the results of 3 random seeds.

**Baselines**  We compare Uni-Mol with multiple baselines, including supervised and pretraining baselines. D-MPNN [53] and AttentiveFP [54] are supervised GNNs methods. N-gram [55], PretrainGNN [22], GROVER [11], GraphMVP [26], MolCLR [12], and GEM [13] are pretraining methods. N-gram embeds the nodes in the graph and assembles them in short walks as the graph representation. Random Forest and XGBoost [56] are used as predictors for downstream tasks.

**Results**  Table 1 and Table 2 show the experiment results of Uni-Mol and competitive baselines, where the best results are marked in bold. Most baseline results are from the paper of GEM, except for the recent works GraphMVP and MolCLR. The results of GraphMVP are from its paper. As MolCLR uses a different data split setting (without considering chirality), we rerun it with the same

Table 2: Uni-Mol performance on molecular property prediction regression tasks

| | Regression (lower is better ↓) | | | | | |
| | RMSE | | | MAE | | |
| Datasets | ESOL | FreeSolv | Lipo | QM7 | QM8 | QM9 |
| # Molecules | 1128 | 642 | 4200 | 6830 | 21786 | 133885 |
| # Tasks | 1 | 1 | 1 | 1 | 12 | 3 |
| D-MPNN | 1.050(0.008) | 2.082(0.082) | 0.683(0.016) | 103.5(8.6) | 0.0190(0.0001) | 0.00814(0.00001) |
| Attentive FP | 0.877(0.029) | 2.073(0.183) | 0.721(0.001) | 72.0(2.7) | 0.0179(0.001) | 0.00812(0.00001) |
| N-Gram$_{RF}$ | 1.074(0.107) | 2.688(0.085) | 0.812(0.028) | 92.8(4.0) | 0.0236(0.0006) | 0.01037(0.00016) |
| N-Gram$_{XGB}$ | 1.083(0.082) | 5.061(0.744) | 2.072(0.030) | 81.9(1.9) | 0.0215(0.0005) | 0.00964(0.00031) |
| PretrainGNN | 1.100(0.006) | 2.764(0.002) | 0.739(0.003) | 113.2(0.6) | 0.0200(0.0001) | 0.00922(0.00004) |
| GROVER$_{base}$ | 0.983(0.090) | 2.176(0.052) | 0.817(0.008) | 94.5(3.8) | 0.0218(0.0004) | 0.00984(0.00055) |
| GROVER$_{large}$ | 0.895(0.017) | 2.272(0.051) | 0.823(0.010) | 92.0(0.9) | 0.0224(0.0003) | 0.00986(0.00025) |
| GraphMVP | 1.029(0.033) | - | 0.681(0.010) | - | - | - |
| MolCLR | 1.271(0.040) | 2.594(0.249) | 0.691(0.004) | 66.8(2.3) | 0.0178(0.0003) | - |
| GEM | 0.798(0.029) | 1.877(0.094) | 0.660(0.008) | 58.9(0.8) | 0.0171(0.0001) | 0.00746(0.00001) |
| Uni-Mol | **0.788(0.029)** | **1.480(0.048)** | **0.603(0.010)** | **41.8(0.2)** | **0.0156(0.0001)** | **0.00467(0.00004)** |

data split setting as other baselines. From the results, we can summarize them as follows: 1) overall, Uni-Mol outperforms baselines on almost all downstream datasets. 2) In solubility (Lipo), free energy (FreeSolv), and quantum mechanical (QM7, QM8, QM9) properties prediction tasks, Uni-Mol is significantly better than baselines. As 3D information is critical in these properties [14; 15], it indicates that Uni-Mol can learn a better 3D representation than other baselines. 3) Uni-Mol fails to beat SOTA on the SIDER dataset. After investigation, we find that Uni-Mol fails to generate 3D conformations for many molecules (like natural products and peptides) in SIDER. Therefore, due to the missing 3D information, it is reasonable that Uni-Mol cannot outperform others.

In summary, by better utilizing 3D information in pretraining, Uni-Mol outperforms all previous MRL models in almost all property prediction tasks.

## 3.2 MOLECULAR CONFORMATION GENERATION

We leave the details of molecular conformation generation to Appendix C.4, as paper [57] pointed out that the current benchmark for molecular conformation generation could be wrong.

## 3.3 POCKET PROPERTY PREDICTION

**Datasets and setup** Druggability, the ability of a candidate protein pocket to produce stable binding to a specific molecular ligand, is one of the most critical properties of a candidate protein pocket. However, this task is very challenging due to the very limited supervised data. For example, NRDLD [58], a commonly used dataset, only contains 113 data samples. Therefore, besides NRDLD, we construct a regression dataset for benchmarking pocket property prediction performance. Specifically, based on Fpocket tool, we calculate Fpocket Score, Druggability Score, Total SASA, and Hydrophobicity Score for the selected 164,586 candidate pockets. The model is finetuned to predict these scores.

**Baselines** On the NRDLD dataset, we compare Uni-Mol with 6 previous methods evaluated in [59]. Accuracy, recall, precision, and F1-score are used as metrics for this classification task. On our created benchmark dataset, as there are no appropriate baselines, we use an additional Uni-Mol model without pretraining, denoted as Uni-Mol$_{no\_pretrained}$, to check the performance brought by pretraining on pocket property prediction.

**Results** As shown in Table 3, Uni-Mol shows the best accuracy, recall, and F1-score on NRDLD. In our created benchmark dataset, the pretraining Uni-Mol model largely outperforms the non-pretraining one on all four scores. This indicates that pretraining on candidate protein pockets indeed brings improvement in pocket property prediction tasks. Unlike Molecular property prediction, due to the very limited supervised data, pocket property prediction gained much less attention. Therefore, we also release our created benchmark dataset, and hopefully, it can help future research.

Table 3: Uni-Mol performance on pocket property prediction

| Dataset | Classification (higher is better ↑) NRDLD | | | | | | Methods | Regression (lower is better ↓) Our Created | |
|---|---|---|---|---|---|---|---|---|---|
| Methods | Cavity-DrugScore | Volsite | DrugPred | PockDrug | TRAPP-CNN | **Uni-Mol** | | Uni-Mol$_{no\_pretrained}$ | **Uni-Mol** |
| Accuracy | 0.82 | 0.89 | 0.89 | 0.865 | 0.946 | **0.973** | RMSE$_{Pocket}$ | 0.1155(0.002) | **0.1140(0.001)** |
| Recall | - | - | - | 0.957 | 0.913 | **1.000** | RMSE$_{Druggability}$ | 0.1117(0.002) | **0.1001(0.001)** |
| Precision | - | - | - | 0.846 | **1.000** | 0.958 | RMSE$_{Total\ SASA}$ | 22.010(0.460) | **20.734(0.015)** |
| F1-score | - | - | - | 0.898 | 0.955 | **0.979** | RMSE$_{Hydrophobicity}$ | 1.4144(0.034) | **1.2847 (0.005)** |

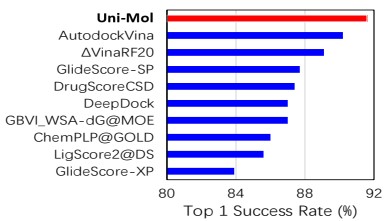

| | Ligand RMSD | | | | |
|---|---|---|---|---|---|
| | % Below Threshold ↑ | | | | |
| Methods | 1.0 Å | 1.5 Å | 2.0 Å | 3.0 Å | 5.0 Å |
| Autodock Vina | 44.21 | 57.54 | 64.56 | 73.68 | 84.56 |
| Vinardo | 41.75 | 57.54 | 62.81 | 69.82 | 76.84 |
| Smina | **47.37** | 59.65 | 65.26 | 74.39 | 82.11 |
| Autodock4 | 21.75 | 31.58 | 35.44 | 47.02 | 64.56 |
| Uni-Mol$_{no\_pretrained}$ | 39.65 | 63.16 | 72.98 | 83.51 | 91.58 |
| Uni-Mol | 43.16 | **68.42** | **80.35** | **87.02** | **94.04** |

Figure 3: Docking power evaluation on CASF-2016 (Top 10 methods)

Table 4: Uni-Mol performance on binding pose prediction

## 3.4 PROTEIN-LIGAND BINDING POSE PREDICTION

**Datasets and setup** As mentioned above, protein-ligand binding pose prediction is one of the most important tasks in drug design. And Uni-Mol combines both the molecular and pocket pretraining models to learn a distance matrix based scoring function, then optimize the complex conformations. For the benchmark dataset, referring to the previous works [28; 60], we use CASF-2016 as the test set. For the training data used in finetuning, we use PDBbind General set v.2020 [61] (19,443 complexes). Notably, to examine the generalization ability, we further filter out the training complexes that are similar to the ones in the test set (CASF-2016). In particular, the complexes with both high protein sequence similarity (MMSeqs2 [62] similarity above 40%) and high molecular similarity (fingerprint similarity above 80%) are filtered out, and there are 18,404 complexes after filtering.

Two benchmarks are conducted: 1) Docking power, the default metric to benchmark the ability of a scoring function in CASF-2016. Specifically, it tests whether a scoring function can distinguish the ground truth binding pose from a set of decoys or not. CASF-2016 provides 50-100 decoy conformations of the same ligand for each ground truth. Scoring functions are applied to rank them, and the ground truth is expected to be the top 1. 2) Binding pose accuracy. Specifically, we use the semi-flexible docking setting: keep the pocket conformation fixed, while the ligand conformation is fully flexible. We evaluate the RMSD between the prediction and the ground truth. Following previous works, we use the percentage of results below predefined RMSD thresholds as metrics.

**Baselines** For the docking power benchmark, the baselines are DeepDock [60] and the top 10 scoring functions reported in [28], including both conventional scoring functions and machine learning-based ones. For the binding pose accuracy, the baselines are Autodock Vina [63; 64], Vinardo [65], Smina [66], and AutoDock4 [67].

**Results** From the docking power benchmark results shown in Figure 3, Uni-Mol ranks the 1st, with the top 1 success rate of 91.2%. For comparison, the previous top scoring function AutoDock Vina [63; 64] achieves 90.2% of the top 1 success rate in this benchmark. From the binding pose accuracy results shown in Table 4, Uni-Mol also outperforms other baselines. Notably, Uni-Mol outperforms the second best method by relatively 22.58% under the threshold of 2Å. This result indicates that Uni-Mol can effectively learn the 3D information from both molecules and pockets, as well as their interaction in the 3D space. Even without pretraining, Uni-Mol (denoted as Uni-Mol$_{random}$) also performs very well. This demonstrates the effectiveness of Uni-Mol backbone, as it effectively learns the 3D information by limited data. In summary, by combining molecular and pocket pretraining models, Uni-Mol significantly outperforms the widely used docking tools in the protein-ligand binding tasks. We leave the efficiency benchmark and visualization for binding pose prediction to Appendix E.

## 4 RELATED WORK

**Representation learning**   In recent years, representation learning [1; 2; 3] has received much attention and has been prevailing in many applications, like in NLP [4; 5; 68; 6; 7], CV [8; 69; 70], or multi-modal[71; 72; 73]. It is not doubted that representation learning becomes a default technology in various tasks.

**Molecular representation learning**   Representation learning on large-scale unlabeled molecules attracts much attention recently. SMILES-BERT [18] is pretrained on SMILES strings of molecules using BERT [4]. Subsequent works are mostly pretraining on 2D molecular topological graphs [23; 11]. MolCLR [12] applies data augmentation to molecular graphs at both node and graph levels, using a self-supervised contrastive learning strategy to learn molecular representations. Further, several recent works try to leverage the 3D spatial information of molecules, and focus on contrastive or transfer learning between 2D topology and 3D geometry of molecules. For example, GraphMVP [26] proposes a contrastive learning GNN-based framework between 2D topology and 3D geometry. GEM [13] uses bond angles and bond length as additional edge attributes to enhance 3D information.

**SE(3)-Equivariant models**   In many-body scenarios such as potential energy surface fitting, SE-(3) equivariance is usually required. A series of SE(3) models are proposed, such as SchNet [74], tensor field networks [30], SE(3) Transformer [31], DimmNet [75], equivariant graph neural networks (EGNN) [41], GemNet [37] and SphereNet [76]. Most of these models are designed for supervised learning with energy and force.

**Pocket druggability prediction**   Druggability prediction of protein binding pockets is crucial for drug discovery as druggable pockets need to be identified at the beginning. Since proteins undergo conformation changes that might alter the druggability of pockets, it is necessary to utilize 3D spatial data beyond sequential information. Early methods, such as Volsite [77], DrugPred [58], and PockDrug [78], predict druggability based on the predefined descriptors of pockets' static structures. Later, TRAPP-CNN [59], based on 3D-CNN, proposes the analysis of proteins' conformation changes and the use of such information for druggability prediction.

**Protein-ligand binding pose prediction**   In structure-based drug design, it is crucial to understand the interactions between protein targets and ligands. The *in vitro* estimation of the binding pose and affinity, such as docking, allows for lead identification and guides molecular optimization. In particular, docking is one of the most important approaches in structure-based drug design and has been developed for the past decades. Tools such as AutoDock4 [67], AutoDock Vina [63; 64], and Smina [66] are among the most used docking programs. Also, machine learning-based docking methods, such as $\Delta_{Vina}RF_{20}$ [79] and DeepDock [60] have also been developed to predict protein-ligand binding poses and assess protein-ligand binding affinity. Equibind [80] is a recent graph deep learning based methods. However, Uni-Mol cannot have an apple-to-apple comparison with Equibind, due to Equibind being proposed for Blind Docking. While Uni-Mol is currently designed for Targeted Docking, which follows most previous traditional tools in docking [81]. The difference is that Blind Docking uses whole protein for docking, while Target Docking directly uses the pocket. We will extend Uni-Mol to Blind Docking tasks in future work.

## 5 CONCLUSION

In this paper, to enlarge the application scope and representation ability of molecular representation learning (MRL), we propose Uni-Mol, the first universal large-scale 3D MRL framework. Uni-Mol consists of 3 parts: a Transformer based backbone to handle 3D data; two large-scale pretraining models to learn molecular and pocket representations respectively; finetuning strategies for all kinds of downstream tasks. Experiments demonstrate that Uni-Mol can outperform existing SOTA in various downstream tasks, especially in 3D spatial tasks.

There are 3 potential future directions. 1) Better interaction mechanisms for finetuning two pretraining models together. As the interaction between the pretraining pocket model and the pretraining molecular model is simple in the current version of Uni-Mol, we believe there is a large room for further improvement. 2) Large Uni-Mol models. As larger pretraining models often perform better, it is worthy of training a large Uni-Mol model on a bigger dataset. 3) More high-quality benchmarks. Although there have been many applications in the field of drug design, high-quality public datasets have been lacking. Many public datasets cannot satisfy real-world demand due to the low data quality. We believe the high-quality benchmarks will be the lighthouse of the entire field, and will significantly accelerate the development of drug design.

ACKNOWLEDGMENTS

We thank Shuqi Lu, Yuanqi Du, Zhen Wang, Yingze Wang, Junhan Chang, Xi Chen, Zhengdan Zhu and many colleagues in DP Technology for their great help in this project. We thank Yean Cheng for his discovery of a bug in the pocket property task.

The work was partially done at Gaoling School of Artificial Intelligence, Peng Cheng Laboratory, Beijing Key Laboratory of Big Data Management and Analysis Methods and MOE Key Lab of Data Engineering and Knowledge Engineering. This research was supported in part by National Natural Science Foundation of China (No. U2241212, No. 61972401, No. 61932001, No. 61832017), by Beijing Natural Science Foundation (No. 4222028), by the major key project of PCL (PCL2021A12), by Beijing Outstanding Young Scientist Program No.BJJWZYJH012019100020098. We also wish to acknowledge the support provided by Engineering Research Center of Next-Generation Intelligent Search and Recommendation, Ministry of Education. Additionally, we acknowledge the support from Intelligent Social Governance Interdisciplinary Platform, Major Innovation & Planning Interdisciplinary Platform for the "Double-First Class" Initiative, Public Policy and Decision-making Research Lab, Public Computing Cloud, Renmin University of China.

3D structures in Fig. 1 are drawn using the web service Hermite™(`https://hermite.dp.tech`).

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

## A   PRETRAINING DATA

**Molecular dataset**   The pretraining datasets we use consist of two parts: one part is a database collection of 12 million molecules that can be synthesized and purchased (See Table 5), and the other part is taken from a previous work [23], whose molecules are collected from the ZINC [82] and ChemBL [83] databases. After normalizing and duplicating, we obtain 19 million molecules as our pretraining dataset. To generate 3D conformations, we use ETKGD [42] with Merck Molecular Force Field [43] optimization in RDKit [44] to randomly generate 11 conformations for each molecule, totally 209M conformations. In these 11 conformations of one molecule, there is a special flattened 3D conformation (atoms with zero z-axis coordinates) that is directly from the molecular graph. This flattened 3D conformation is used for the cases where RDKit failed to generate 3D conformations, like the peptides in the SIDER task.

**Candidate protein pocket dataset**   The pretraining dataset for candidate protein pockets is derived from the Protein Data Bank (RCSB PDB [2]) [45], a collection of 180K structural data of proteins. We first pre-process the raw data by adding missing side chains and hydrogen atoms, and then we use Fpocket [46] to detect candidate binding pockets of the proteins. After filtering the raw pockets by the number of residues they have contact with (10~25) and including water molecules inside the pockets, we collect a pretraining dataset of 3,291,739 candidate pockets.

## B   DOWNSTREAM DATA SUPPLEMENTS

### B.1   MOLECULAR PROPERTY PREDICTION

We conduct experiments on the MoleculeNet[52] benchmark in the molecular property prediction task. MoleculeNet is a widely used benchmark for molecular property prediction. The details of the 15 datasets we used are described below.

- **BBBP** Blood-brain barrier penetration (BBBP) contains the ability of small molecules to penetrate the blood-brain barrier.

- **BACE** This dataset contains the results of small molecules as inhibitors of binding to human $\beta$-secretase 1 (BACE-1).

- **ClinTox** This dataset contains the toxicity of the drug in clinical trials and the status of the drug for FDA approval[84].

- **Tox21** The dataset contains toxicity measurements of 8k molecules for 12 targets.

- **ToxCast** This dataset is derived from toxicology data from in vitro high-throughput screening and contains toxicity measurements for 8k molecules against 617 targets.

- **SIDER** The Side Effect Resource (SIDER) contains side effects of drugs on 27 system organs. These drugs are not only small molecules but also some peptides with molecular weights over 1000.

- **HIV** This dataset contains 40k compounds with the ability to inhibit HIV replication.

- **PCBA** PubChem BioAssay (PCBA) is a database of small molecule bioactivities generated by high-throughput screening. This is a subset containing over 400k molecules on 128 bioassays.

- **MUV** Maximum Unbiased Validation (MUV) is another subset of PubChem BioAssay, containing 90k molecules and 17 bioassays.

- **ESOL** This dataset contains the water solubility of the compound and is a small dataset with 1128 molecules.

- **FreeSolv** The dataset contains hydration free energy data for small molecules, of which we use the experimental values as labels.

- **Lipo** Lipophilicity contains the solubility of small molecules in lipids, of which we use the octanol/water distribution coefficient as the label.

---

[2]http://www.rcsb.org/

Table 5: Database collection of 12M purchasable molecules

| Database | Molecules | Link |
|---|---:|---|
| Targetmol | 10,000 | `https://www.targetmol.com/` |
| Chemdiv | 1,613,931 | `https://www.chemdiv.com/` |
| Enamine | 2,734,581 | `https://enamine.net/` |
| Chembridge | 1,557,942 | `https://www.chembridge.com/` |
| Life Chemical | 509,975 | `https://lifechemicals.com/` |
| Specs | 208,670 | `https://www.specs.net/` |
| Vitas-M | 1,409,339 | `https://vitasmlab.biz/` |
| InterBioScreen | 48,627 | `https://www.ibscreen.com/` |
| Maybridge | 53,352 | `https://www.thermofisher.in/` |
| Bionet-Key Organics | 259,244 | `https://www.keyorganics.net/` |
| Asinex | 530,881 | `https://www.asinex.com/` |
| UkrOrgSynthesis | 688,952 | `https://uorsy.com/` |
| Eximed | 61,009 | `https://eximedlab.com/` |
| HTS Biochemie Innovationen | 58,437 | `https://www.hts-biochemie.de/` |
| Princeton BioMolecular | 1,532,542 | `https://princetonbio.com/` |
| Otava | 270,835 | `https://otavachemicals.com/` |
| Alinda Chemical | 202,332 | `https://www.alinda.ru/` |
| Analyticon | 42,664 | `https://www.analyticon-diagnostics.com/` |

- **QM7, QM8, QM9** The molecule in QM7 contains up to 7 heavy atoms, QM8 is 8 and QM9 is 9. These datasets provide the geometric, energetic, electronic and thermodynamic properties of the molecule, which are calculated by density functional theory (DFT)[85]. QM9 contains several quantum mechanical properties of different quantitative ranges, and we select *homo*, *lumo* and *gap* of similar quantitative range, following the setup of the previous work[13].

## B.2 Molecular corformation generation

Following the settings in previous works [49; 86], we use GEOM-QM9 and GEOM-Drugs [87] dataset in this task.

- **GEOM** This dataset contains 37 million accurate conformations generated for 450,000 molecules by advanced sampling and semi-empirical density functional theory (DFT). Of these, 133,000 molecules are from QM9, and the remaining 317,000 molecules have biophysical, physiological, or physical chemistry experimental data, i.e., Drugs.

## B.3 Pocket property prediction

NRDLD [58] is a benchmark dataset for pocket druggability prediction. As NRDLD and other existing benchmark datasets are too small, we construct a regression dataset to benchmark pocket property prediction performance.

- **NRDLD** The dataset contains 113 proteins, and a predefined split is provided: 76 proteins constitute the training set and 37 proteins constitute the test set. It labels 71 proteins as druggable in that they noncovalently bind small drug-like ligands [59]. The rest 42 proteins are labeled as less-druggable because none of the ligands they cocrystallized satisfy the following requirements simultaneously: the rule of five, clogP $\geq$ -2, and ligand efficiency, as defined in [29], $\geq 0.3$ kcal mol$^{-1}$ / heavy atom.

- **Our created benchmark dataset** The dataset contains 164,586 candidate pockets, and four scores (Fpocket Score, Druggability Score, Total SASA, and Hydrophobicity Score) calculated by the Fpocket tool. These four scores are indicators of the druggability of candidate pockets. To avoid leaking, the selected pockets are not overlapped with the candidate protein pocket dataset used in Uni-Mol pretraining.

Table 6: Uni-Mol hyperparameters setup during pre-training

| Hyperparameter | Molecular pretraining | Pocket pretraining |
|---|---|---|
| Layers | 15 | 15 |
| Peak learning rate | 1e-4 | 1e-4 |
| Batch size | 128 | 128 |
| Max training steps | 1M | 1M |
| Warmup steps | 10K | 10k |
| Attention heads | 64 | 64 |
| FFN dropout | 0.1 | 0.1 |
| Attention dropout | 0.1 | 0.1 |
| Embedding dropout | 0.1 | 0.1 |
| Weight decay | 1e-4 | 1e-4 |
| Embedding dim | 512 | 512 |
| FFN hidden dim | 2048 | 2048 |
| Gaussian kernel channels | 128 | 128 |
| Corrupt ratio | 0.15 | 0.15 |
| Activation function | GELU | GELU |
| Learning rate decay | Linear | Linear |
| Adams $\epsilon$ | 1e-6 | 1e-6 |
| Adams $(\beta_1, \beta_2)$ | (0.9, 0.99) | (0.9, 0.99) |
| Gradient clip norm | 1.0 | 1.0 |
| Atom loss function and its weight | Cross entropy, 1.0 | Cross entropy, 1.0 |
| Coordinate loss function and its weight | Smooth L1, 5.0 | Smooth L1, 1.0 |
| Distance loss function and its weight | Smooth L1, 10.0 | Smooth L1, 1.0 |
| Vocabulary size (atom types) | 30 | 9 |

### B.4 PROTEIN-LIGAND BINDING POSE PREDICTION

We use PDBbind General set v.2020 [61], excluding the similar complexes in CASF-2016 [28], as the training set. And CASF-2016 is used as the test set. In particular, we define the pocket for each protein-ligand pair as residues of the protein which have at least one atom within the range of 6Å from a heavy atom in the ligand. All atoms of the selected residues are included. In addition, we draw the smallest bounding box covering all of the atoms in the pocket and regard the water molecules in the bounding box as a part of the pockets, too.

- **PDBbind General set v.2020** This dataset contains 19,443 protein-ligand complexes with binding data and processed structural files originally from the Protein Data Bank (PDB). Only complexes with experimentally determined binding affinity data are included in the general set. Notably, to examine the generalization ability, we further filter out the training complexes that are similar to the ones in the test set (CASF-2016). In particular, the complexes with both high protein sequence similarity (MMSeqs2 [62] similarity above 40%) and high molecular similarity (fingerprint similarity above 80%) are filtered out, and there are 18,404 complexes after filtering.

- **CASF-2016** CASF-2016 is the widely used benchmark for docking and scoring. This dataset, whose primary test set is known as the PDBbind Core set, contains 285 protein-ligand complexes with high-quality crystal structures and reliable binding constants from PDBbind General set. For each protein-ligand complex, CASF-2016 provides 50~100 decoy molecular conformations of the same ligand for evaluation.

## C EXPERIMENTS DETAILS & REPRODUCE

### C.1 MOLECULAR PRETRAINING SETUP

We report the detailed hyperparameters setup of Uni-mol during pretraining in Table 6. Uni-Mol training loss is summed up by three components, atom (token) loss, coordinate loss, and pair-distance loss. Atoms are masked, and noise is added to coordinate as described in sections 2.1 and 2.2. Since the values of the above three components differ significantly, to make them have a similar influence, we enlarge the coordinate loss and distance loss. Molecular pretraining runs on 8 V100 GPUs (32GB memory, the same below), and the training time is about 20 hours.

Table 7: Search space for small datasets: BBBP, BACE, ClinTox, Tox21, Toxcast, SIDER, ESOL, FreeSolv, Lipo, QM7, QM8, for large datasets: PCBA, MUV, QM9, and for HIV

| Hyperparameter | Small | Large | HIV |
|---|---|---|---|
| Learning rate | [5e-5, 8e-5, 1e-4, 4e-4, 5e-4] | [2e-5, 1e-4] | [2e-5, 5e-5] |
| Batch size | [32, 64, 128, 256] | [128, 256] | [128, 256] |
| Epochs | [40 ,60, 80, 100] | [20, 40] | [2, 5, 10] |
| Pooler dropout | [0.0, 0.1, 0.2, 0.5] | [0.0, 0.1] | [0.0, 0.2] |
| Warmup ratio | [0.0, 0.06, 0.1] | [0.0, 0.06] | [0.0, 0.1] |

## C.2 POCKET PRETRAINING SETUP

The pocket Uni-Mol model is slightly different from molecule ones during pretraining: 1) We use a residue-level masking strategy instead of the original atom-level, as residue granularity is non-redundancy and integrity in protein. 2) Hydrogen is removed in pocket Uni-Mol pretraining, to reduce the number of used atoms and thus improve efficiency. 3) All weights of loss functions are set 1, as the residue-level masking strategy makes the 3D denoising task much harder. 4) A different vocabulary is used in pocket pretraining. In pocket data, there are amino acids, whose atoms are mostly C, N, O, S and H. While in molecule data, the atom types are more diverse, so a larger vocabulary is used. Other settings are listed in Table 6. Pocket pretraining runs on 8 V100 GPUs, and the training time is about 2 days and 20 hours.

## C.3 MOLECULAR PROPERTY PREDICTION

- **Data split** In our experiments, referring to previous work GEM[13], we use scaffold splitting[88] to divide the dataset into training, validation, and test sets in the ratio of 8:1:1. Scaffold splitting is more challenging than random splitting as the scaffold sets of molecules in different subsets do not intersect. This splitting tests the model's generalization ability and reflects the realistic cases[52]. Since this splitting is according to the scaffold of the molecule, we find that whether or not chirality is considered when generating the scaffold using RDKit has a significant impact on the division results. From the results, the splitting considering chirality makes the task harder. The original implementation of MolCLR does not consider chirality, and we reproduce the experiment by considering it. In all experiments, we choose the checkpoint with the best validation loss, and report the results on the test-set run by that checkpoint.

- **Hyperparameter search space** Referring to previous works, we use a grid search to find the best combination of hyperparameters for the molecular property prediction task. To reduce the time cost, we set a smaller search space for the large datasets. The specific search space is shown in Table 7. For small datasets, we run them on a single V100 GPU; for large datasets and HIV, we run them on 4 V100 GPUs.

## C.4 MOLECULAR CONFORMATION GENERATION

- **Datasets and setup** Following the settings in previous works [49; 86], we use GEOM-QM9 and GEOM-Drugs [87] dataset to perform conformation generation experiments. As described in Sec. 2.3, in this task, Uni-Mol optimizes its input generative conformations to the labeled ones. To construct the finetuning data, we leverage RDKit to generate input conformations. For each input, we calculate the RMSD between it and labeled conformations, and choose the one with minimal RMSD as its optimizing target. For the inference in the test set, we generate the same number of conformations (twice the number of labeled conformations) as previous works do. And we also use the same metrics, Coverage (COV) and Matching (MAT) as in previous works. Higher COV means better diversity, while lower MAT means higher accuracy.

- **Metrics** In this task, following previous work [89; 90], we use the Root of Mean Squared Deviations (RMSD) of heavy atoms to evaluate the difference between the generated conformation and the reference one. Before computing RMSD, the generated conformation is first aligned with the reference one, and the function $\Phi$ aligns conformations by applying rotations and translations to them:

$$\text{RMSD}(\boldsymbol{R}, \hat{\boldsymbol{R}}) = \min_{\Phi} (\frac{1}{n} \sum_{i=1}^{n} ||\Phi(\boldsymbol{R}_i) - \hat{\boldsymbol{R}}_i||^2)^{\frac{1}{2}} \quad (4)$$

where $\boldsymbol{R}$ and $\hat{\boldsymbol{R}}$ are the generated and reference conformation, $i$ is the $i$-th heavy atom, and $n$ is the number of heavy atoms.

We use Coverage (COV) and Matching (MAT) to evaluate the performance of the conformation generation model. Higher COV means better diversity, while lower MAT means higher accuracy. Formally, COV and MAT are denoted as:

$$\text{COV}(S_g, S_r) = \frac{\left| \left\{ \boldsymbol{R} \in S_r | \text{RMSD}(\boldsymbol{R}, \hat{\boldsymbol{R}}) < \delta, \hat{\boldsymbol{R}} \in S_g \right\} \right|}{|S_r|} \tag{5}$$

$$\text{MAT}(S_g, S_r) = \frac{1}{|S_r|} \sum_{\boldsymbol{R} \in S_r} \min_{\hat{\boldsymbol{R}} \in S_g} \text{RMSD}(\boldsymbol{R}, \hat{\boldsymbol{R}}) \tag{6}$$

where $S_g$ and $S_r$ are the set of generated and reference conformations, respectively, and $\delta$ is a given RMSD threshold. Following previous work [49; 86], for GEOM-QM9, the threshold is 0.5Å, and for GEOM-Drugs, the threshold value is 1.25Å.

- **Baselines** We compare Uni-Mol with 10 competitive baselines. RDKit [42] is a traditional conformation generation method based on distance geometry. The rest baseline can be categorized into two classes. GraphDG [48], CGCF[49], ConfVAE [91], ConfGF [86], and DGSM [92] combine generative models with distance geometry, which first generates interatomic distance matrices and then iteratively generates atomic coordinates. CVGAE [50], GeoMol [51], DMCG [93], and GeoDiff [94] directly generate atomic coordinates.

- **Results** The results are shown in Table 8. We report the mean and median of COV and MAT on GEOM-QM9 and GEOM-Drugs datasets. ConfVAE [91] and DGSM's [92] results are from their papers, respectively. GeoMol[51] results are from DMCG [93] paper, which DMCG reproduced after aligning the data split. We found the test sets used in GeoDiff [94] and DMCG [93] are slightly different from baselines (different filtering conditions), so we use the released model parameters of GeoDiff to reproduce the results on the same test set. For DMCG, as model parameters are not released, we use its open-source codes to reproduce from scratch. Other baseline results are from ConfGF's paper. As shown in Table 8, Uni-Mol exceeds existing baselines in both COV and MAT metrics on both datasets.

- **Experiments details** We report the detailed hyperparameters setup for molecular conformation generation in Table 9. And we run this task on a single V100 GPU. Since this is a 3D-related task, we only use coordinate loss and distance loss. Following AlphaFold [38], we use "recycling" to iteratively refine the output atom positions. We leverage RDKit (ETKGD) for generating inputs in molecular conformation generation tasks. Specifically, in finetuning, we randomly generate up to 2000 conformations and cluster them into 10 conformations, as the model input. A similar pipeline is used in the inference of test data. For most baselines, as they aim to generate conformations from scratch, RDKit-generated conformations are not leveraged.

## C.5 POCKET PROPERTY PREDICTION

The hyperparameters we search are listed in Table 10. We run this task on a single V100 GPU. When we use the Fpocket [46] tool in our experiments, it outputs four values: Fpocket score, Druggability score, Total SASA, and Hydrophobicity Score. Specifically, the Fpocket score is a custom score by Fpocket; the druggability score is an empirical score calculated from evolution and homologous information. Besides, to verify the effectiveness of the Fpocket tool on real-world data, we test this tool on NRDLD. Table11 shows the performance of Fpocket tool on NRDLD dataset.

## C.6 PROTEIN-LIGAND BINDING POSE PREDICTION

- **Data split** The training set is PDBbind General set v.2020 excluding the similar complexes covered by CASF-2016. We perform data preprocessing, such as adding missing atoms to both proteins and ligands and manually fixing file-loading errors, before constructing the training set. And we filter the complexes based on the number of residues contained in the pockets (>= 5 ). Besides, we filter similar complexes in PDBbind with CASF-2016 in consideration of protein sequence and ligand similarity. MMseqs2 [62] similarity above 40% and RDKit FingerprintSimilarity [44] above 80% are used as filtering conditions. The test set is CASF-2016, which contains 285 protein-ligand complexes.

Table 8: Uni-Mol performance on molecular conformation generation. Please note the previous conformation generation models were not pretrained.

| Dataset | QM9 | | | | Drugs | | | |
|---|---|---|---|---|---|---|---|---|
| | COV(↑, %) | | MAT(↓, Å) | | COV(↑, %) | | MAT(↓, Å) | |
| Methods | Mean | Median | Mean | Median | Mean | Median | Mean | Median |
| RDKit | 83.26 | 90.78 | 0.3447 | 0.2935 | 60.91 | 65.70 | 1.2026 | 1.1252 |
| CVGAE | 0.09 | 0.00 | 1.6713 | 1.6088 | 0.00 | 0.00 | 3.0702 | 2.9937 |
| GraphDG | 73.33 | 84.21 | 0.4245 | 0.3973 | 8.27 | 0.00 | 1.9722 | 1.9845 |
| CGCF | 78.05 | 82.48 | 0.4219 | 0.3900 | 53.96 | 57.06 | 1.2487 | 1.2247 |
| ConfVAE | 80.42 | 85.31 | 0.4066 | 0.3891 | 53.14 | 53.98 | 1.2392 | 1.2447 |
| ConfGF | 88.49 | 94.13 | 0.2673 | 0.2685 | 62.15 | 70.93 | 1.1629 | 1.1596 |
| GeoMol | 71.26 | 72.00 | 0.3731 | 0.3731 | 67.16 | 71.71 | 1.0875 | 1.0586 |
| DGSM | 91.49 | 95.92 | 0.2139 | 0.2137 | 78.73 | 94.39 | 1.0154 | 0.9980 |
| GeoDiff | 92.65 | 95.75 | 0.2016 | 0.2006 | 88.45 | 97.09 | 0.8651 | 0.8598 |
| DMCG | 94.98 | 98.47 | 0.2365 | 0.2312 | 91.27 | 100.00 | 0.8287 | 0.7908 |
| Uni-Mol$_{no\_pretrained}$ | 97.00 | 100.00 | 0.1907 | 0.1754 | 91.68 | 100.00 | 0.8102 | 0.8041 |
| Uni-Mol | **97.95** | **100.00** | **0.1831** | **0.1659** | **91.91** | **100.00** | **0.7863** | **0.7794** |

Table 9: Hyperparameters setup for molecular conformation generation

| Hyperparameter | QM9 | Drugs |
|---|---|---|
| Learning rate | 2e-5 | 5e-5 |
| Batch size | 128 | 128 |
| Epochs | 50 | 70 |
| Warmup ratio | 0.06 | 0.06 |
| Recycling times | 4 | 4 |
| Coordinate loss function and weight | MSE, 1.0 | MSE, 1.0 |
| Distance loss function and weight | MSE, 1.0 | MSE, 1.0 |

- **Model architecture and finetuning** As shown in Figure 4, the binding pose model is an encoder-decoder architecture consisting of two 15-layer Uni-Mol as encoders and a 4-layer Uni-Mol as a decoder. The decoder block follows the same setting as the pretraining ones. During finetuning, Uni-Mol is trained towards to cross distances (between pocket and ligand) and holo distances (pair distances in ligand itself) with MSE loss. The hyper-parameters used in fine-tuning are listed in Table 12. The Dist_threshold is introduced to alleviate the large MSE loss from long-range atom pairs and thus stabilize the training. In particular, the atom pairs with distances larger than Dist_threshold are ignored in the loss calculation. Following AlphaFold [38], we use "recycling" to iteratively refine the output atom positions. We finetune model on 4 V100 GPUs.

- **Binding pose prediction** As aforementioned, Uni-Mol does not directly predict the binding pose in finetuning. We use a scoring function based method to optimize the input coordinates. We first construct a scoring function by the difference between the current pair-distance matrix and the predicted pair-distance matrix. Given the coordinates of pocket atoms and ligand atoms, we can compute its pair-distance matrix. We use the mean square error to get the difference between the current pair-distance matrix and the predicted one from the model. Notably, as listed in Table 12, different weights are used in cross distances and holo distances in calculating mean square error. Then, to get the binding pose, we directly apply back-propagation from the scoring function to

Table 10: Search space for pocket property prediction

| Hyperparameter | NRDLD | Our Created |
|---|---|---|
| Learning rate | [5e-5, 1e-4, 3e-4] | 3e-4 |
| Batch size | [1, 2, 4, 8, 16] | 32 |
| Epochs | 40 | 20 |
| Pooler dropout | [0, 0.1, 0.2, 0.3] | 0 |
| Warmup ratio | [0.0, 0.1] | 0.1 |

Table 11: Performance of Fpocket tool on NRDLD

|  | Accuracy | Recall | Precision | F1-score |
|---|---|---|---|---|
| Fpocket Score | 0.73 | 0.83 | 0.76 | 0.79 |
| Druggability Score | 0.78 | 0.83 | 0.83 | 0.83 |

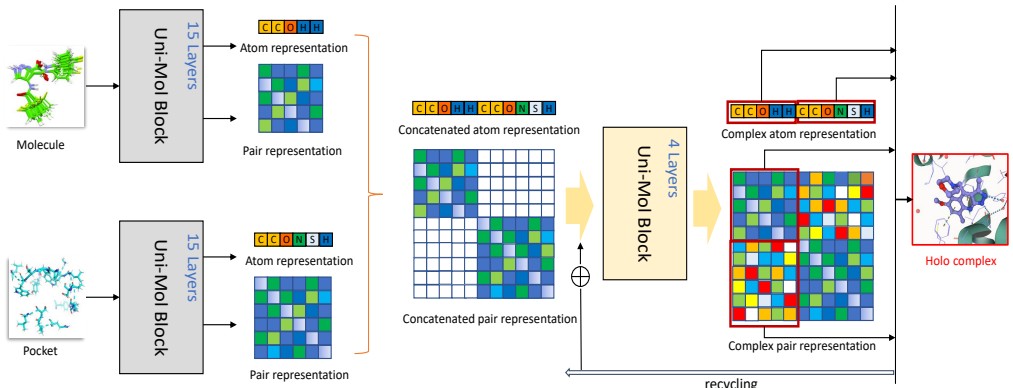

Figure 4: protein-ligand binding pose model: 1) Encoder: molecular representation and pocket representation are obtained from their own pretraining Uni-Mol models; 2) Decoder: representation is concatenated with atom and pair-level, as inputs of a 4 layers Uni-Mol block learning from scratch with recycling. 3) Output: The complex representation is used as a project layer to learn the pair distances of molecule and pocket.

input coordinates. We choose to use LBGFS [95] with a learning rate of 1.0 as the optimizer for efficiency. We use 10 randomly generated molecular conformations with randomly placed positions (6 degrees of freedom) as data augmentation, and choose the one with the lowest loss after back-propagation as the final result.

- **Exhaustiveness search for docking tools** To ensure that the comparison between Uni-Mol and popular molecular docking software is unbiased, we increase the exhaustiveness of the global search (roughly proportional to time) of the molecular docking software to observe the effect of computational complexity on docking power on CASF-2016 benchmark. As shown in Table 13, when exhaustiveness is above 16, the popular molecular docking software can no longer improve the performance by increasing the computational complexity.

## D ABLATION STUDY

### D.1 3D SPATIAL POSITIONAL ENCODINGS BENCHMARK

We investigate the performance of different 3D spatial positional encodings on the 3D molecular pretraining. In particular, we benchmarked:

- **Gaussian kernel (GK)**, a simply Gaussian density function.
- **Gaussian kennel with pair type (GKPT)** [39]. Based on GK, an affine transformation according to the pair type is applied on pair distances, before applying the Gaussian kernel.
- **Radial Bessel basis (RBB)** [75]. A Bessel based radial function.
- **Discretization categorical embedding (DCE)**. We convert the continued distances to the discrete bins, by Discretization. With binned distances, embedding-based positional encoding is directly used.
- **Delta coordinate (DC)** [36]. Following Point Transformer [36], the deltas of coordinates are directly used as pair-wise spatial relative positional encoding.

Table 12: Hyperparameters setup for binding pose prediction

| Hyperparameters for finetuning | Value |
|---|---|
| Learning rate | 3e-4 |
| Batch size | 32 |
| Epochs | 50 |
| Warmup ratio | 0.06 |
| Dropout | 0.2 |
| Dist_threshold | 8.0 |
| Recycling times | 3 |
| Cross distance loss function and weight | MSE, 1.0 |
| Holo distance loss function and weight | MSE, 1.0 |
| Hyperparameters for binding pose prediction | Value |
| Optimizer | LBFGS |
| Learning rate | 1.0 |
| Max iterations | 100 |
| Early stopping | 5 |
| Dist_threshold | 4.5 |
| Conformation size | 10 |
| Cross distance weight | 1.0 |
| Holo distance weight | 5.0 |

Table 13: Exhaustiveness study of popular docking tools on CASF-2016

| | | Ligand RMSD | | | |
| | | % Below Threshold ↑ | | | |
| Methods | Exhaustiveness | 0.5 Å | 1.0 Å | 1.5 Å | 2.0 Å |
|---|---|---|---|---|---|
| Autodock Vina | 1 | 21.40 | 35.79 | 47.02 | 52.28 |
| Autodock Vina | 8 | 23.86 | 44.21 | 57.54 | 64.56 |
| Autodock Vina | 16 | 25.61 | 45.96 | 60.70 | 66.67 |
| Autodock Vina | 32 | 25.96 | 45.96 | 60.00 | 66.32 |
| Vinardo | 1 | 16.84 | 33.33 | 43.16 | 49.82 |
| Vinardo | 8 | 23.51 | 41.75 | 57.54 | 62.81 |
| Vinardo | 16 | 23.51 | 45.26 | 60.70 | 66.67 |
| Vinardo | 32 | 23.86 | 44.56 | 59.30 | 65.61 |
| Smina | 1 | 23.51 | 39.65 | 50.53 | 56.14 |
| Smina | 8 | 23.51 | 47.37 | 59.65 | 65.26 |
| Smina | 16 | **28.77** | 49.47 | 61.40 | **67.72** |
| Smina | 32 | 28.07 | **51.23** | **61.75** | 67.37 |
| Autodock4 | 1 | 4.91 | 18.95 | 26.67 | 28.87 |
| Autodock4 | 8 | 7.02 | 21.75 | 31.58 | 35.44 |
| Autodock4 | 16 | 6.32 | 24.56 | 34.04 | 38.95 |
| Autodock4 | 32 | 6.32 | 23.16 | 34.04 | 38.25 |

- **Gaussian kennel with pair type and local graph (GKPTLG)**. Based on GKPT, we set up a model with locally connected graphs. In particular, the cutoff radius is set to 6 Å.

We summarize the validation loss during pretraining for them in Fig. 5. From the results, we can find:

- The performance of DCE and GK are almost the same, and outperform RBB and DC. And we choose GK as the basic encoding.

- Compared with GK, GKPT convergences faster. This indicates the pair type is critical in the 3D spatial positional encoding.

- Compared with GKPT, GKPTLG convergences slower. This indicates the locally cutoff graph is not effective for self-supervised learning, and the default fully connected graph in Transformer is more effective.

- As GKPT outperforms all other encodings, we use it in the backbone model of Uni-Mol.

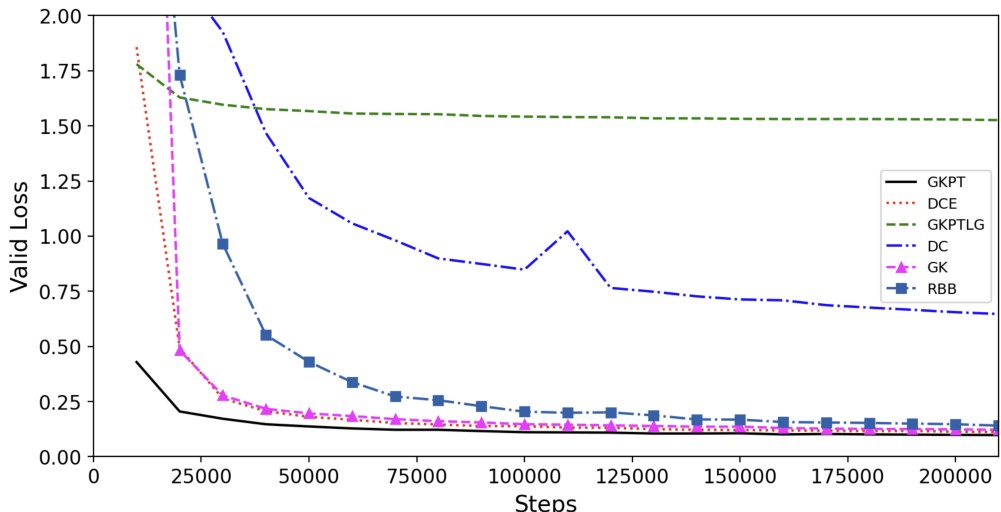

Figure 5: Validation loss in pretraining for different 3D spatial encodings

Table 14: Ablation studies, molecular property prediction classification tasks

| | BBBP | BACE | ClinTox | Tox21 | ToxCast | SIDER | HIV | PCBA | MUV |
|---|---|---|---|---|---|---|---|---|---|
| | Classification (ROC-AUC %, higher is better ↑) | | | | | | | | |
| Datasets | BBBP | BACE | ClinTox | Tox21 | ToxCast | SIDER | HIV | PCBA | MUV |
| Uni-Mol w/o pretraining | 69.0(0.7) | 80.9(5.4) | 68.3(2.2) | 75.8(0.4) | 63.8(0.1) | 61.9(0.5) | 76.2(2.4) | 86.1(0.5) | 62.8(4.0) |
| Uni-Mol w/o pair representation | 71.6(1.3) | 85.4(2.7) | 85.5(1.7) | 79.4(0.1) | 69.3(0.1) | 64.3(0.9) | 80.2(0.2) | 88.4(0.1) | 71.0(7.7) |
| 2D shortest path encoding | 71.6(2.1) | 85.6(1.1) | 83.6(4.0) | 79.6(0.7) | 68.8(0.8) | 63.7(0.1) | 78.9(0.4) | 88.0(0.2) | 78.2(0.6) |
| 1D relative positional encoding | 70.3(1.9) | 77.8(3.7) | 64.2(2.0) | 73.3(0.7) | 64.9(0.2) | 61.5(1.6) | 75.6(0.3) | 77.2(1.4) | 68.7(1.0) |
| Uni-Mol | **72.9(0.6)** | **85.7(0.2)** | **91.9(1.8)** | **79.6(0.5)** | **69.6(0.1)** | **65.9(1.3)** | **80.8(0.3)** | **88.5(0.1)** | **82.1(1.3)** |

Table 15: Ablation studies, molecular property prediction regression tasks

| | ESOL | FreeSolv | Lipo | QM7 | QM8 | QM9 |
|---|---|---|---|---|---|---|
| | Regression (lower is better ↓) | | | | | |
| | RMSE | | | MAE | | |
| Datasets | ESOL | FreeSolv | Lipo | QM7 | QM8 | QM9 |
| Uni-Mol w/o pretraining | 0.924(0.037) | 1.880(0.206) | 0.745(0.012) | 45.2(0.6) | 0.0174(0.0002) | 0.00653(0.00040) |
| Uni-Mol w/o pair representation | 0.807(0.027) | 1.582(0.068) | 0.611(0.004) | 45.2(1.0) | 0.0158(0.0001) | 0.00573(0.00004) |
| 2D shortest path encoding | 0.831(0.007) | 1.482(0.070) | 0.605(0.003) | 60.6(0.2) | 0.0164(0.0001) | 0.00650(0.00001) |
| 1D relative positional encoding | 0.929(0.035) | 2.237(0.074) | 0.866(0.004) | 77.5(2.7) | 0.0283(0.0007) | 0.02283(0.00078) |
| Uni-Mol | **0.788(0.029)** | **1.480(0.048)** | **0.603(0.010)** | **41.8(0.2)** | **0.0156(0.0001)** | **0.00467(0.00004)** |

## D.2    Pair representation benchmark

Pair representation is a new component proposed in Uni-Mol for enhancing 3D spatial positional encodings. To check its effectiveness, we conduct an ablation benchmark. We add a setting that only uses spatial positional encoding as attention bias in attention layers, and the attention weights of the last layer are used for the coordinate update and distance prediction. The downstream molecular property prediction results, shown in Tables 14 and 15, demonstrate that the one without pair representation is worse. Besides, according to the efficiency benchmark in Table 17, we can find the extra cost introduced by pair representation is very small. This result demonstrates the efficiency of the proposed pair representation.

## D.3    SE(3) coordinate head benchmark

As aforementioned in Sec. 2.1, in SE(3) coordinate head, we use delta pair representation to calculate the coordinate updates, while previous work EGNN [41] did not. Here we conduct a benchmark on molecular conformation generation. As shown in Table 16, Uni-Mol's SE(3) head outperforms the one in EGNN. The result indicates the effectiveness of the proposed SE(3) coordinate head.

Table 16: Benchmark on SE(3)-equivariant coordinate head.

| Dataset | QM9 | | | | Drugs | | | |
|---|---|---|---|---|---|---|---|---|
| Methods | COV($\uparrow$, %) | | MAT($\downarrow$, Å) | | COV($\uparrow$, %) | | MAT($\downarrow$, Å) | |
| | Mean | Median | Mean | Median | Mean | Median | Mean | Median |
| Uni-Mol w/ EGNN head | 96.94 | 100.00 | 0.1873 | 0.1730 | **91.93** | 98.80 | 0.7952 | 0.7821 |
| **Uni-Mol** | **97.95** | **100.00** | **0.1831** | **0.1659** | 91.91 | **100.00** | **0.7863** | **0.7794** |

## D.4    Efficiency benchmark

Due to the massive pretraining cost in the large-scale data, we favor the efficient backbone model. And the backbone model in Uni-Mol is designed based on the principle. To demonstrate the efficiency of our designed backbone, we compared it with vanilla Transformer and SE(3)-Transformer [31] in terms of speed. We use the same benchmark dataset, QM9, in SE(3)-Transformer NVIDIA version [3], to test the training speed, and summarize the results in Table 17. We can see that Uni-Mol backbone model is only slightly slower than the vanilla Transformer and much faster than SE(3)-Transformer(and NVIDIA optimized version). The result explicitly demonstrates that our design backbone model is indeed very efficient.

Table 17: Efficiency benchmark, we test vanilla Transformer and add our components based on it. Besides, SE(3) Transformer [31] is also benchmarked. "seconds" is the average cost seconds for a batch. "ratio" is the slowdown ratio compared with vanilla Transformer.

| Methods | Params. | seconds | ratio |
|---|---|---|---|
| vanilla Transformer | 47.57M | 0.062 | 1.00 |
| vanilla Transformer + spatial position encoding | 47.59M | 0.063 | 1.02 |
| vanilla Transformer + spatial position encoding + pair repr. | 47.60M | 0.064 | 1.03 |
| Uni-Mol | 47.61M | 0.066 | 1.06 |
| SE(3)-Transformer$_{public}$ [31] | 46.77M | 14.776 | 238.32 |
| SE(3)-Transformer$_{nvidia}$ | 47.02M | 0.388 | 6.26 |

## D.5    Pretraining strategies benchmark

We investigate and benchmark several pretraining strategies with Uni-Mol framework.

- **Masked autoencoders (MAE)**. Following MAE [96], the uncorrupted atoms are used as input of the encoder, while corrupted atoms are not. Then a decoder accepts both uncorrupted and corrupted atoms as input. The MAE is proven effective in image-related tasks.

---

[3]https://github.com/NVIDIA/DeepLearningExamples/tree/master/DGLPyTorch/DrugDiscovery/SE3Transformer

- **Masked atom prediction only (MAP)**. This setting only uses masked atom prediction, without 3D position recovery.

- **MAP+3D**. The setting in Uni-Mol, with MAP and 3D position recovery.

- **MAP+3D+CL(Contrastive Learning)**. A contrastive loss [97; 98] is further added for the 3D conformations. In particular, two conformations are treated as "positive" if they are from the same molecule; otherwise, they are treated as "negative." The loss is combined together with 3D position recovery and masked atom prediction tasks.

Besides the above strategies, we also add two additional methods. 1) Single Conformation. This method only uses one 3D conformation for each molecule, to examine the performance brought by more 3D conformations. 2) Local-Connected Graph. This method uses the local-connected graph, by radius cutoff 6 Å, to examine the performance brought by the fully connected graph in Transformer.

The results are shown in Table 18, and we can find the pretraining strategies in Uni-Mol (MAP+3D) is very effective, outperforming the other settings.

From the result of Single Conformation, we can find that more 3D conformations can slightly boost the performance of the molecular property prediction tasks. As for the results of the Local-Connected Graph, we can find the performance brought by the fully connected graph in Transformer is quite significant.

Table 18: Pretraining strategies benchmark

| | ROC-AUC %, higher is better ↑ | | | | RMSE↓ | | | MAE↓ |
|---|---|---|---|---|---|---|---|---|
| | BBBP | BACE | ClinTox | SIDER | ESOL | FreeSolv | Lipo | QM7 |
| MAE | 71.7(0.6) | 86.4(0.6) | 82.7(3.2) | 63.8(0.4) | 0.851(0.008) | 1.586(0.034) | 0.605(0.008) | 45.7(0.7) |
| MAP | 71.3(0.3) | 86.1(0.5) | 90.2(2.8) | 62.5(1.1) | 0.835(0.015) | 1.562(0.035) | 0.595(0.010) | 46.6(0.2) |
| MAP+3D (Uni-Mol) | 72.9(0.6) | 85.7(0.2) | **91.9(1.8)** | **65.9(1.3)** | **0.788(0.029)** | 1.480(0.048) | 0.603(0.010) | **41.8(0.2)** |
| MAP+3D+CL | 72.4(0.9) | 84.2(0.6) | 88.8(0.7) | 65.2(0.9) | 0.846(0.026) | 1.499(0.020) | 0.604(0.004) | 44.6(2.7) |
| Single Conformation | **73.6(0.7)** | 85.8(1.0) | 91.3(2.8) | 65.6(0.6) | 0.796(0.022) | **1.479(0.051)** | 0.607(0.007) | 43.7(0.8) |
| Local-Connected Graph | 70.2(1.1) | **87.5(0.6)** | 87.5(3.8) | 64.2(1.7) | 0.826(0.029) | 1.580(0.050) | **0.593(0.011)** | 47.3(1.2) |

### D.6 Corrupted Position Generation and Assignment

To generate corrupted positions for corrupted atoms, we proposed Alg.1, which first generates random positions based on ground-truth ones, plus with noises; then, an optional permutational re-assignment is used to further reduce the delta positions. Since there is a tradeoff between noise range and re-assignment, we benchmarked several settings, and list the result in Table 19.

From the result, we can find:

- "$r$=1Å" is better than "$r$=1Å, re-assign". This indicates, when $r$ is small, the re-assignment is not needed.

- "$r$=1.5Å, re-assign" is better than "$r$=1.5Å". This indicates, when $r$ is large, the re-assignment is needed.

- when $r$ is large, e.g. 2Å, the final performance is not good.

- "$r$=1Å" performs best in our benchmark, and we use it as our final setting for corrupted position generation.

Table 19: Corrupted Position Generation and Assignment

| | ROC-AUC %, higher is better ↑ | | | | RMSE↓ | | | MAE↓ |
|---|---|---|---|---|---|---|---|---|
| | BBBP | BACE | ClinTox | SIDER | ESOL | FreeSolv | Lipo | QM7 |
| $r$=2Å, re-assign | 70.7(1.1) | 84.9(1.4) | 87.4(0.8) | 65.1(1.2) | 0.827(0.016) | 1.557(0.013) | 0.601(0.017) | 43.1(0.5) |
| $r$=1.5Å, re-assign | 72.2(0.3) | 85.0(1.3) | 90.4(3.9) | 64.6(0.5) | 0.826(0.019) | 1.526(0.072) | 0.599(0.007) | 43.8(1.1) |
| $r$=1.5Å | 71.7(0.9) | 85.0(1.5) | 88.3(3.4) | 64.4(1.2) | 0.819(0.030) | 1.484(0.072) | 0.588(0.011) | 43.8(0.7) |
| $r$=1Å, re-assign | 72.1(3.2) | 86.5(1.2) | 88.0(0.8) | 64.6(2.0) | 0.825(0.012) | 1.613(0.092) | 0.600(0.008) | 44.9(0.2) |
| $r$=1Å | 72.9(0.6) | 85.7(0.2) | 91.9(1.8) | 65.9(1.3) | 0.788(0.029) | 1.480(0.048) | 0.603(0.010) | 41.8(0.2) |

## D.7 PRETRAINING OR NOT

From Tables 14 and 15, it is clear that pretraining can significantly boost the downstream performance.

## D.8 1D/2D POSITIONAL ENCODINGS

We also investigate the impact of 3D spatial positional encoding on molecular property prediction tasks. Specifically, to demonstrate the effectiveness of introducing 3D information, we replace the original invariant spatial position encoding with a 2D Graphormer-like[24] shortest path positional encoding and a 1D BERT-like[4] relative positional encoding on atoms. The results are summarized in the following table. Tables 14 and 15 show the results of the ablation studies. We can find that 3D spatial positional encoding largely improves the performance of molecular property prediction. It is clear that 3D information indeed helps the performance of downstream tasks.

## E    MORE RESULTS FOR BINDING POSE PREDICTION

**Efficiency benchmark**    We compare Uni-Mol binding pose prediction with popular docking tools in efficiency. The baseline results are taken from EquiBind [99] paper.And Uni-Mol binding pose prediction is run on a single V100 GPU. For each molecule, Uni-Mol is run with 10 different initial conformations, and the total time cost is reported. As shown in Table 20, Uni-Mol is significantly faster than traditional docking tools, about 250x faster.

Table 20: Efficiency on binding pose prediction. The average seconds per ligand are reported.

| Methods | QVINA-W | GNINA | SMINA | GLIDE (C.) | Uni-Mol |
|---|---|---|---|---|---|
| seconds | 49 | 247 | 146 | 1405 | **0.2** |

**Visualization**    We show protein-ligand binding pose prediction in CASF-2016 test dataset in Figure 6. Green molecules are the Uni-Mol predictions while red ones are the ground truth in complexes. From the Figure, we can find that Uni-Mol can predict the accurate binding complexes, with large overlapping with ground-truth ligands.

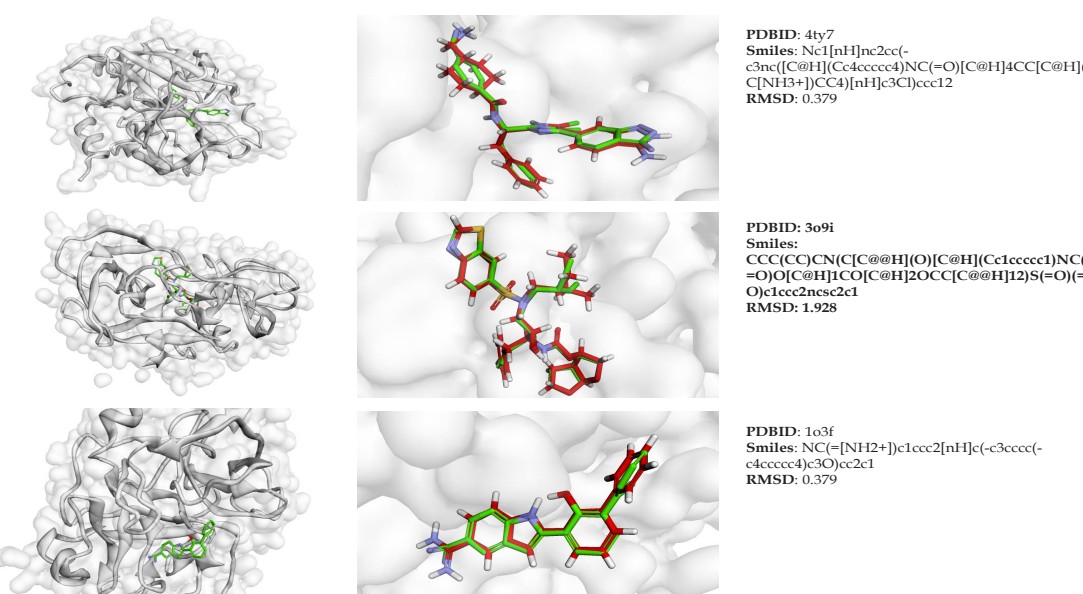

PDBID: 4ty7
**Smiles**: Nc1[nH]nc2cc(-c3nc([C@H](Cc4ccccc4)NC(=O)[C@H]4CC[C@H](C[NH3+])CC4)[nH]c3Cl)ccc12
**RMSD**: 0.379

PDBID: 3o9i
**Smiles**: CCC(CC)CN(C[C@@H](O)[C@H](Cc1ccccc1)NC(=O)O[C@H]1CO[C@H]2OCC[C@@H]12)S(=O)(=O)c1ccc2ncsc2c1
**RMSD**: 1.928

PDBID: 1o3f
**Smiles**: NC(=[NH2+])c1ccc2[nH]c(-c3cccc(-c4ccccc4)c3O)cc2c1
**RMSD**: 0.379

Figure 6: Binding pose prediction visualization in CASF-2016

**Ligand conformation performance**    As shown in Table 4, Uni-Mol cannot outperform popular docking tools in RMSD below 1.0Å. After investigation, we found it is due to the RMSD of ligand itself is not good, as shown in Table 21. We hypothesize that the binding pose prediction in Uni-Mol is not end-to-end, and gradient descent based optimization is not physics/chemical aware, so it may generate inaccurate ligand conformations. In contrast, popular docking tools are mostly physics/chemical aware, like sampling ligand conformations by enumerating rotatable bonds [65]. To tackle this, a simple workaround is to combine the Uni-Mol and docking tools: Uni-Mol mostly focuses on binding positions, while docking tools focus on ligand conformation. Considering the physics/chemical constraints and predicting binding pose end-to-end are also worthy to try. We leave the further optimizations as future work.

Table 21: Ligand-only conformation performance on CASF-2016

| | Ligand RMSD % Below Threshold ↑ | | |
|---|---|---|---|
| Methods | 0.5 Å | 1.0 Å | 2.0 Å |
| Uni-Mol$_{random}$ | 14.39 | 57.54 | 89.82 |
| Uni-Mol | 17.20 | 57.19 | 93.33 |

## F    TRAINING STABILITY

With Pre-LayerNorm [32] backbone and mixed-precision training, the pretraining sometimes diverges. After investigation, we found there are large numerical values in the intermediate states when divergence happens. We hypothesize that the Final-LayerNorm layer in the Pre-LayerNorm backbone results in the problem. Specifically, Final-LayerNorm is applied to the sum of all encoder layers, denoted as

$$\boldsymbol{o}_i = \text{LayerNorm}(\boldsymbol{s}_i), \quad \boldsymbol{s}_i = \sum_{l=1}^{L} \boldsymbol{o}_i^l \tag{7}$$

where $L$ is the number of layers, $\boldsymbol{o}_i^l$ is the output of the $i$-th position in the $l$-th layer, and $\boldsymbol{o}_i$ is the final output of the $i$-th position, after Final-LayerNorm. Therefore, due to normalization, $\boldsymbol{s}_i$ can be arbitrarily large (or arbitrarily small), without affecting model results. However, a too large or too small numerical value will cause the numerical unstable, especially in the mixed-precision training. To tackle this, we introduce a simple loss, to restrict the value range of $\boldsymbol{s}_i$. Formally, the loss is denoted as

$$\mathcal{L}_{norm} = \text{mean}_i \left( \max \left( \left| \|\boldsymbol{s}_i\| - \sqrt{d} \right| - \tau, 0 \right) \right), \tag{8}$$

where $d$ is the dimension size of $\boldsymbol{s}_i$, $\tau$ is the tolerance factor. In Uni-Mol, we set $\tau = 1$, and both atom-level and pair-level representations are constrained by this loss. Besides, to avoid affecting other loss functions, we set a very small loss weight (0.01) to $\mathcal{L}_{norm}$.

## G    SELF-ATTENTION MAP VISUALIZATION

For better interpretability, we conduct a visualization on the self-attention map and pair distance of the molecule, shown in the Figure 7. We can easily find the average weight of all attention heads (in the right-most figures) is very similar to the distance matrix. However, when we check these heads independently, there are different patterns, and are different with distance matrix. For example, most attention weights are asymmetric, while the distance matrix is symmetry. Besides, the long-range interaction could be captured by the attention (like in Head 11 and 13). The learned long-range interaction patterns reflect our motivation for using Transformer as the backbone model.

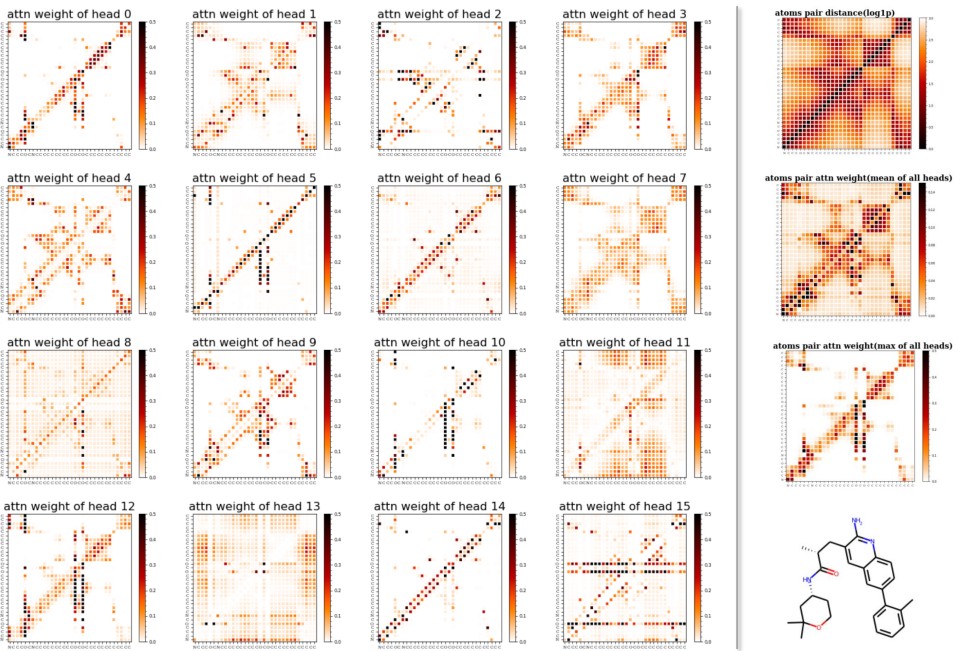

Figure 7: Visualization on self-attention map of multi heads independently.

# H  VISUALIZATION FOR PRETRAINING TASKS.

We visualize 3D position recovery ability in Uni-Mol pretraining process. As shown in Figure 8, Uni-Mol recovers coordinates and atoms with almost no mistakes in conformational space.

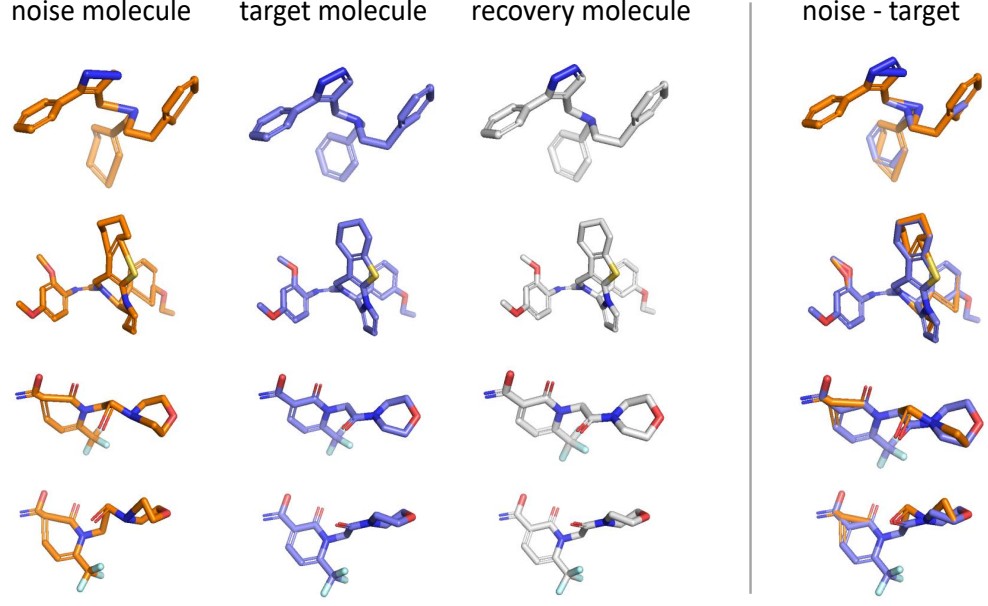

Figure 8: Visualization of 3D position recovery in pretraining.

Moreover, we plot the pretraining masked atom predictions accuracy and the coordinate recovery loss of the validation set. As shown in Figure 9, during pretraining, Uni-Mol can accurately predict the atom type of the masked atoms and recover the corrupted 3D positions.

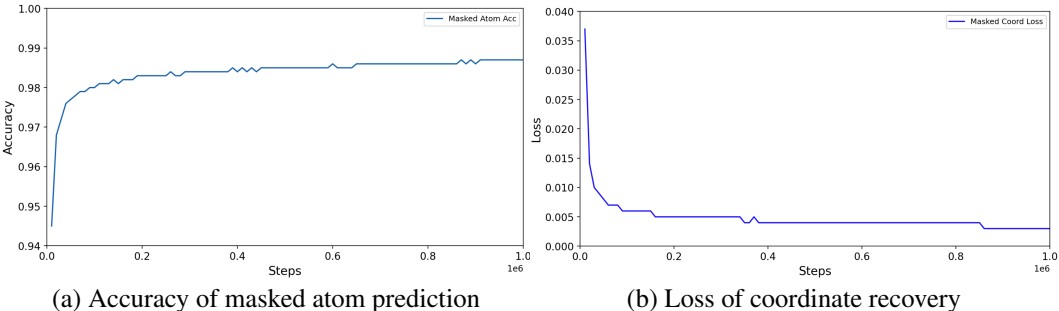

(a) Accuracy of masked atom prediction          (b) Loss of coordinate recovery

Figure 9: Validation losses in pretraining.

## I   COMPARISON WITH GRAPHORMER AND 3D-GRAPHORMER

The positional encoding (the shortest path) used in Grahpormer can only handle 2D molecular graphs, not 3D positions. In the Uni-Mol backbone, we added several modifications to make the model have the ability to handle 3D inputs and outputs. Further, there is a concurrent following-up work called 3D-Graphormer [35], adapting Graphormer to 3D molecules. There are several differences between us: 1) Both Uni-Mol and 3D-Graphormer use the pair-wise Euclidean distance and Gaussian kernel to encode 3D spatial information. 3D-Graphormer has an additional node-level centrality encoding, which is the sum of spatial encodings of each node. 2) 3D-Graphormer doesn't have pair representation. 3) Our SE(3) Coordinate Head is different from the "node-level projection head" in 3D-Graphormer. The method used in 3D-Graphormer is an attention layer for 3 axes in 3D coordinates. 4) 3D-Graphormer is not designed for self-supervised pretraining.

## J   ONE TABLE FOR MOLECULAR PROPERTY PREDICTION

This section is one clear table 22 to associate the pretraining dataset information and the corresponding results from Table 1 and Table 2 for molecular property prediction.

## K   NEURIPS 2022 REVIEW COMMENTS

The Neurips 2022 review comments of Uni-Mol can be found at `https://openreview.net/forum?id=IfFZr1gl0b` for reference.

Table 22: Uni-Mol performance on molecular property prediction tasks, with information of pretraining datasets. "Z", "C", "P", "G", and "U" in the Data Sources column denote the used pretraining data, which are the abbreviation of ZINC15 [82], ChEMBL [83], PubChem [100], GEOM [87] and Uni-Mol (described in Appendix A). "*" denotes unsupervised pretraining on the corresponding downstream dataset. The ZINC data is a subset of ZINC15, and different models may use different sample sizes. Different data sources usually have redundancy, such as ZINC and ChemBL, and need to be de-duplicated when used.

| Datasets | Classification (ROC-AUC %, higher is better ↑) | | | | | | | | | Regression (lower is better ↓) | | | | | | Data | |
| | | | | | | | | | | RMSE | | | MAE | | | | |
| | BBBP | BACE | ClinTox | Tox21 | ToxCast | SIDER | HIV | PCBA | MUV | ESOL | FreeSolv | Lipo | QM7 | QM8 | QM9 | # Pretraining Molecules (M) | Data Sources |
| # Molecules | 2039 | 1513 | 1478 | 7831 | 8575 | 1427 | 41127 | 437929 | 93087 | 1128 | 642 | 4200 | 6830 | 21786 | 133885 | | |
| # Tasks | 1 | 1 | 2 | 12 | 617 | 27 | 1 | 128 | 17 | 1 | 1 | 1 | 1 | 12 | 3 | | |
| D-MPNN | 71.0(0.3) | 80.9(0.6) | 90.6(0.6) | 75.9(0.7) | 65.5(0.3) | 57.0(0.7) | 77.1(0.5) | 86.2(0.1) | 78.6(1.4) | 1.050(0.008) | 2.082(0.082) | 0.683(0.016) | 103.5(8.6) | 0.0190(0.0001) | 0.00814(0.00001) | no pretrain | - |
| Attentive FP | 64.3(1.8) | 78.4(0.022) | 84.7(0.3) | 76.1(0.5) | 63.7(0.2) | 60.6(3.2) | 75.7(1.4) | 80.1(1.4) | 76.6(1.5) | 0.877(0.029) | 2.073(0.183) | 0.721(0.001) | 72.0(2.7) | 0.0179(0.001) | 0.00812(0.00001) | no pretrain | - |
| N-Gram$_{RF}$ | 69.7(0.6) | 77.9(1.5) | 77.5(4.0) | 74.3(0.4) | - | 66.8(0.7) | 77.2(0.1) | - | 76.9(0.7) | 1.074(0.107) | 2.688(0.085) | 0.812(0.028) | 92.8(4.0) | 0.0236(0.0006) | 0.01037(0.00016) | * | - |
| N-Gram$_{XGB}$ | 69.1(0.8) | 79.1(1.3) | 87.5(2.7) | 75.8(0.9) | - | 65.5(0.7) | 78.7(0.4) | - | 74.8(0.2) | 1.083(0.082) | 5.061(0.744) | 2.072(0.030) | 81.9(1.9) | 0.0215(0.0005) | 0.00964(0.00031) | * | - |
| PretrainGNN | 68.7(1.3) | 84.5(0.7) | 72.6(1.5) | 78.1(0.6) | 65.7(0.6) | 62.7(0.8) | 79.9(0.7) | 86.0(0.1) | 81.3(2.1) | 1.100(0.006) | 2.764(0.002) | 0.739(0.003) | 113.2(0.6) | 0.0200(0.0001) | 0.00922(0.00004) | 2.46 | Z+C |
| GROVER$_{base}$ | 70.0(0.1) | 82.6(0.7) | 81.2(3.0) | 74.3(0.1) | 65.4(0.4) | 64.8(0.6) | 62.5(0.9) | 76.5(2.1) | 67.3(1.8) | 0.983(0.090) | 2.176(0.052) | 0.817(0.008) | 94.5(3.8) | 0.0218(0.0004) | 0.00984(0.00055) | 11.00 | Z+C |
| GROVER$_{large}$ | 69.5(0.1) | 81.0(1.4) | 76.2(3.7) | 73.5(0.1) | 65.3(0.5) | 65.4(0.1) | 68.2(1.1) | 83.0(0.4) | 67.3(1.8) | 0.983(0.090) | 2.176(0.052) | 0.817(0.008) | 94.5(3.8) | 0.0218(0.0004) | 0.00984(0.00055) | 11.00 | Z+C |
| GraphMVP | 72.4(1.6) | 81.2(0.9) | 79.1(2.8) | 75.9(0.5) | 63.1(0.4) | 63.9(1.2) | 77.0(1.2) | - | 77.7(0.6) | 1.029(0.033) | - | 0.681(0.010) | - | - | - | 0.05 | G |
| MolCLR | 72.2(2.1) | 82.4(0.9) | 91.2(3.5) | 75.0(0.2) | - | 58.9(1.4) | 78.1(0.5) | - | 79.6(1.9) | 1.271(0.040) | 2.594(0.249) | 0.691(0.004) | 66.8(2.3) | 0.0178(0.0003) | - | 10.00 | P |
| GEM | 72.4(0.4) | 85.6(1.1) | 90.1(1.3) | 78.1(0.1) | 69.2(0.4) | 67.2(0.4) | 80.6(0.9) | 86.6(0.1) | 81.7(0.5) | 0.798(0.029) | 1.877(0.094) | 0.660(0.008) | 58.9(0.8) | 0.0171(0.0001) | 0.00746(0.00001) | 20.00 | Z |
| Uni-Mol | **72.9(0.6)** | **85.7(0.2)** | **91.9(1.8)** | **79.6(0.5)** | **69.6(0.1)** | 65.9(1.3) | **80.8(0.3)** | **88.5(0.1)** | **82.1(1.3)** | **0.788(0.029)** | **1.480(0.048)** | **0.603(0.010)** | **41.8(0.2)** | **0.0156(0.0001)** | **0.00467(0.00004)** | 19.00 | U |

