# OpenReview forum: "Uni-Mol: A Universal 3D Molecular Representation Learning Framework"
_ICLR.cc/2023/Conference — ICLR 2023 poster_

### Official Review · Reviewer_LVDL · 2022-10-19

**Confidence:** 4
**Correctness:** 2
**Technical Novelty And Significance:** 2
**Empirical Novelty And Significance:** 2
**Recommendation:** 5

**Clarity, Quality, Novelty And Reproducibility:**

The paper is well-written. The method is clearly presented.  The detailed appendix and promised code release enabled reproducibility. However, the novelty is questionable.

**Strength And Weaknesses:**

## Strength

#### 1. This paper motivates using unlabeled data to improve molecular or protein representation learning via self-supervised learning or pertaining tasks, which is a popular topic in the research area.

#### 2. The two pretraining tasks are technically sound.

#### 3. Two large-scale datasets are collected for pretraining.

#### 4. The paper outperforms most existing methods.

#### 5. The code, model, and data will be made publicly available.


## Weaknesses

#### 1. I thought the novelty is questionable.  The authors claimed that the proposed Uni-Mol is the first pure 3D molecular pretraining framework. However, there have been already a few similar works. For example,

a. The Graph Multi-View Pre-training (GraphMVP)  framework leverages the correspondence and consistency between 2D topological structures and 3D geometric views.

Liu et al., Pre-training Molecular Graph Representation with 3D Geometry, ICLR 2021.

b. The geometry-enhanced molecular representation learning method (GEM)  proposes includes several dedicated geometry-level self-supervised learning strategies to learn molecular geometry knowledge.

Fang et al., Geometry-enhanced molecular representation learning for property prediction, nature machine intelligence, 2022.


c. Guo et al. proposed a self-supervised pre-training model for learning structure embeddings from protein 3D structures.

Guo et al., Self-Supervised Pre-training for Protein Embeddings Using Tertiary Structures, AAAI 2022.

d. The GeomEtry-Aware Relational Graph Neural Network (GearNet) framework uses type prediction, distance prediction and angle prediction of masked parts for pretaining.

Zhang et al., Protein Representation Learning by Geometric Structure Pretraining, ICML 2022 workshop.

#### 2. The comparison with the SOTA methods may be unfair. The performance of the paper is based on the newly collected 209M dataset. However, the existing methods use smaller datasets. For example, GEM employs only 20M unlabeled data. Because the scale of datasets has a significant impact on the accuracy, the superior of the proposed method may be from the new large-scale datasets.

#### 3. The authors claimed one of the contributions is that the proposed Uni-Mol contains a simple and efficient SE(3)-equivariant Transformer backbone. However, I thought this contribution is too weak.

#### 4. The improvement is not very impressive or convincing. Although with a larger dataset for pretraining,  the improvement is a bit limited, e.g., in Table 1.

#### 5. It is not clear which part causes the main improvement: Transformer, pretraining or the larger dataset?

#### 6. It could be better to show the  3D position recovery and masked atom prediction accuracy and visualize the results.

#### 7. The visualization of the self-attention map and pair distance map in Appendix H is interesting. However, according to the visualization, the self-attention map is very similar to the pair distance map, as the author explained. In this case, why not directly use pair distance as attention? Or what does self-attention actually learn besides distance in the task? As self-attention is computationally expensive, is it really needed?

**Summary Of The Paper:**

This paper proposes a self-supervised or pretrained method for molecular representation learning. It can be summarized as follows,

#### 1. A Transformer-based backbone.

#### 2. Two self-supervised learning or pretraining tasks: 3D position recovery and masked atom prediction.

#### 3. A few finetuning strategies are presented for fine-tuning or downstream tasks.

**Summary Of The Review:**

The paper presents a  pretraining framework for molecular representation learning. I like the newly collected dataset, which may be potentially useful to the topic.

However, the novelty may be overclaimed. For a pretraining work, the most important thing is the self-learning design. However, predicting the property or attributes of the masked or missing parts is straightforward and has been widely used in existing works.  The experimental comparison is also questionable.

---

> ### Public Comment · ~Karlsson_Yu1 · 2022-11-10
> **Regarding reviewer LVDL's Weaknesses comment 3**
>
> Dear reviewer LVDL, authors and readers,
>
> I am an independent researcher actively working in this area, and I believe the SE(3) equivariant head proposed in Uni-Mol is in fact quite novel and enlightening. Its novelty mainly lies in its simplicity. This is quite clear when one compares this equivariant head to the one that has been proposed in GeoDiff (equations 5, 6,7 in GeoDiff paper). GeoDiff uses an equivariant network whose backbone is a graph neural network. As a result, it needs multiple layers of message passing to adequately propagate information among atoms for updating the coordinates in an SE(3) equivariant way. I also notice that this equivariant GNN is needed in every diffusion step of GeoDiff. Considering GeoDiff needs 5000 diffusion steps per molecule (as mentioned here https://openreview.net/forum?id=PzcvxEMzvQC&noteId=6LSfTeOqv9), the computation resource allocated to the equivariant network is not trivial.
>
> In contrast, Uni-Mol uses a Transformer backbone which can be considered as a fully-connected GNN. This greatly expands its information aggregation radius. As a result, it may only need a single self-attention layer to achieve a more effective information aggregation. If I understand it correctly, **Equation (3) reconstructs the corrupted input coordinates back to a valid conformation in a single step using $Q^L$ that is obtained with a single pass of the network**. This might justify Uni-Mol's claim of being more efficient.
>
> **However, I do agree with reviewer LVDL that the comparison is not fair, especially the comparison with other methods in conformation generation experiments in Table 3 where NONE OF THE compared method has pre-training.** All of them are only trained on a much smaller dataset (200,000 conformations I believe), whereas **Uni-Mol is first pre-trained on 209M conformations".
>
> Perhaps, it would be better to also show Uni-Mol performance on conformation generation (with a confidence interval?) without any pre-training. I believe it can substantially consolidate the effectiveness of the simple equivariant head, and further prove superiority of the proposed feature extraction backbone.
>
> I really enjoyed reading this paper and thank you for putting together a concise and comprehensive framework!!

---

> > ### Author Response · Authors · 2022-11-10
> > **Thanks for the comments!**
> >
> > Dear Karlsson,
> >
> > Thank you very much for supporting our work. For your concern about the fair comparison, please refer to our response to the reviewers.
> > In table 3, indeed the molecular generation baselines are not pretrained, and Uni-Mol is the first attempt to use pretraining in the molecular conformation generation task, as far as we know. And we would like to add an experiment that uses Uni-Mol without pretraining as you suggest.

---

> > > ### Author Response · Authors · 2022-11-11
> > > **Additional experiment on molecular conformation generation.**
> > >
> > > We have updated the corresponding result in Table 20 of the paper. From the result, we can find that: 1) Even without pretraining, Uni-Mol still can outperform the previous competitive baselines.  2) With pretraining, the performance could be further improved.
> > >
> > > Like the previous work, we do not list the confidence interval. Actually, the metrics (COV and MAT) are already calculated based on a set of generated results, so the confidence interval is not necessary.

---

> ### Author Response · Authors · 2022-11-10
> **Response to Reviewer LVDL (part 1)**
>
> We thank the reviewer so much for the careful review and insightful comments! It seems the reviewer mostly concerned about the performance of molecular property prediction. We first want to highlight, that Uni-Mol is not just designed for molecular property prediction like most previous MRL frameworks. The goal of Uni-Mol is to **enlarge the application scope** in the drug design field, especially the 3D tasks. And as shown in the experiment section, besides molecular property prediction, Uni-Mol can be used in **molecular conformation generation, pocket property prediction, and protein-ligand complex prediction**. All these tasks are important in drug design, and previous MRL frameworks cannot be used in them directly. Uni-Mol is the first MRL framework that can be used directly in these tasks, and outperforms previous SOTA. We hope the reviewer can recognize this contribution.
>
> - About the first pure 3D molecular pretraining framework (Weakness 1)
>   - Sorry for the misleading, we have revised the contribution to "Uni-Mol is the first pure 3D molecular pretraining framework that can **predict 3D positions**, and the first molecular
> pretraining framework that can be directly **used in 3D tasks in the field of drug design**."  What we want to highlight is the ability to predict/generate 3D positions and to use in 3D tasks in drug design.
>   - As we discussed in Sec.1 and Sec.4, the first two papers (GraphMVP and GEM) you provided are not pure 3D molecular pretraining frameworks. In downstream tasks, both of them cannot take 3D positions as inputs, since the 3D information is only used as an auxiliary loss during pretraining. Besides, they cannot predict 3D positions directly. In Uni-Mol, both the input and output can be 3D positions, in both pretraining and downstream tasks.
>   - The last two papers (Guo et al. and Zhang et al.) you provided are protein structure pretraining frameworks, which are less related to Uni-Mol. In these two papers, although the 3D information is used as input, they still cannot directly predict the 3D positions like Uni-Mol. The self-supervised task in Guo et al. is to recover the distance matrix; Zhang et al. use contrastive learning for the pretraining. Besides, their downstream tasks are not 3D prediction tasks.
>
> - About the size of pretraining datasets (Weakness 2)
>   - As described in Sec. 2.2, the number of molecules used in Uni-Mol is **19M**, the same scale as GEM (20M). Based on the 19M molecules, we generate 11 conformations for each molecule, in total 209M conformations. As a molecule has several different conformations naturally, using multiple conformations is a reasonable data augmentation in 3D MRL.
>   - GEM also uses 10 conformations for each molecule, **in total 200M conformations**. refer to their code:  https://github.com/PaddlePaddle/PaddleHelix/blob/ee1acdb6b3bb6d6cb9cf8c27373669e5c79605c0/pahelix/utils/compound_tools.py#L674 and https://github.com/PaddlePaddle/PaddleHelix/blob/ee1acdb6b3bb6d6cb9cf8c27373669e5c79605c0/pahelix/featurizers/gem_featurizer.py#L213
>   - Combining the above, the comparison with GEM (200M conformations) is fair.
>   - We also conduct an additional experiment that only uses 1 conformation for each molecule, referring to Table 18 in the paper. The result is only slightly worse than the one with 11 conformations, and still outperforms GEM in most tasks.
>
> - About the contribution of SE(3)-equivariant Transformer backbone is too weak (weakness 3)
>   - We updated the contribution to "Based on extensive benchmarks, we build a simple and efficient SE(3)-equivariant Transformer backbone, and an effective 3D pretraining strategy in Uni-Mol."
>   - A simple and effective solution may look easy in its final form, but discovering it is usually not easy. We conduct extensive benchmarks (see Appendix E) to find the simple and effective backbone model and pretraining strategies.
>   - Besides, we believe efficiency is very important in pretraining. Due to the large-scale unlabeled dataset and the large models, the cost of pretraining is very massive. With an efficient model, we can 1) train the model with the same performance at a smaller cost; 2) train a better model (with more data or more model parameters) at the same cost.

---

> > ### Author Response · Authors · 2022-11-10
> > **Response to Reviewer LVDL (part 2)**
> >
> > - About the performance in molecular property prediction (Weakness 4)
> >   - As aforementioned, uni-mol is not just proposed for the molecular property prediction task. In **molecular conformation generation, pocket property prediction and protein-ligand complex prediction**, uni-mol can be directly used, outperforming previous SOTA.
> >   - Referring to "**About the size of pretraining datasets**", the comparison with the previous baseline is fair. Besides, even with fewer data (19M conformations), Uni-Mol still outperforms GEM, which uses 200M conformations.
> >   - We also refer to our previous review comment at NeurIPS (see the supplementary materials), *"molecular property prediction has already been studied for long, and the baseline models are very competitive."*, from Reviewer ByGu.
> >
> > - Which part brings improvement, Transformer, pretraining or the larger dataset? (Weakness 5)
> >   - Referring to "**About the size of pretraining datasets**", the comparison with the previous baseline is fair. Uni-Mol does not use a larger dataset. But we still conduct an experiment that only uses 1 conformation for 1 molecule, in total 19M conformations. As shown in Table 18, the improvement brought by more conformations is marginal.
> >   - As shown in Tables 14 and 15, the pretraining contributes a lot to the molecular property prediction tasks.
> >   - As shown in Table 18, if we use the radius-cutoff local-connected graphs, like in GNN, the performance is also worse. This indicates the fully connected graphs in Transformer also contribute to the improvement.
> >
> > - More visualization study (Weakness 6)
> >   - We add some visualizations of the results of the 3D position recovery task in Figure 8.
> >   - We add the curves of the pretraining tasks in Figure 9.
> >
> > - About visualization of the self-attention map and distance matrix (Weakness 7)
> >   - Sorry for the misleading. The previous visualization of the self-attention map is the average of all attention heads, so it shows an average pattern, which is very close to the distance matrix.
> >   - We update Figure 7, and visualize the weights of different self-attention heads. We can easily find that the patterns in different heads are quite different. And they are also different from the distance matrix.
> >   - In these patterns, there are some long-range interactions ( like "attn head of 13" ). And most of them are asymmetric patterns. The symmetric distance matrix cannot simply replace them.
> >   - Therefore, as the distance matrix cannot represent these different patterns in multi-head attention, it is hard to replace self-attention with the distance matrix.
> >
> > - About "However, the novelty may be overclaimed.  ..." in the Summary of the Review
> >   - Note that any pre-trained models will be used for downstream tasks. Therefore, we believe the most important criterion to evaluate a self-learning design is using downstream task performance. From this point of view, our proposed Uni-Mol is significant as it outperformed the previous SOTA on many popularly used benchmarks. Furthermore,  Uni-Mol is the first self-learning method that works well in 3D tasks in drug design, which demonstrates its wide adaptability and practical value.
> >   - The review further mentioned that masked prediction tasks had been widely used. It is true for CV, NLP, and 1D/2D MRL. However, we would like to point out that, as far as we know, most previous works in 3D MRL focus on contrastive learning or distance/angle prediction (as an auxiliary loss). See the related work section for detail. That being said, the 3D position recovery task (described in Sec. 2.2 and Algorithm 1) used in Uni-Mol is the first attempt to study how to leverage the wisdom in CV/NLP to 3D MRL. The comparison benchmark in Appendix E.5 also demonstrated the effectiveness of the proposed 3D position recovery. However, as the evaluation of novelty is somehow subjective, we didn't list the novelty of the pretraining task as one of our contributions in the paper.
> >
> > We appreciate the reviewer for spending time reviewing our paper and offering valuable suggestions. We sincerely hope that our responses address your concerns and you can reevaluate our submission. We are also willing to discuss with you if you have any further questions.

---

> > > ### Comment · Reviewer_LVDL · 2022-11-15
> > > **Thank you for the reseponses.**
> > >
> > > Thanks for the detailed responses.  It is nice to see the proposed method outperforms existing methods without any pretraining (Table 3). I tend to increase my score but I still have a few questions.
> > >
> > > For the mask-based pretraining approach, first using it in MRL is not very impressive. The authors may remove it from the claimed contributions and organize the paper as a new Transformer-based backbone with the widely-used mask-based pretraining approach for 3D MRL. For experiments,  what I really want to see is a clear table where the results are with the same training data and settings as the competitors and the data information is clearly included in the table.  Last, only comparing Uni-Mol with GEM is not very convincing. More comparisons could be better.
> > >
> > > The motivation of Transformer. It is still not clear what self-attention actually learns. I understand in AlphaFold there is attention for the interaction between MSAs and pair representations.  Is there a significant difference with those AlphaFold-based methods? What do you expect self-attention to learn in Uni-Mol, distance,  similarity or a kind of relation?  Because the visualization shows that self-attention is uninterpretable and messy, I think a distance-based matrix with a few small  Gaussian noises may be also effective.  Moreover, in the first submission, the visualization of self-attention is mainly about distance. However, in the revision, it becomes quite different, which makes me feel the paper is a bit unsolid.

---

> > > > ### Author Response · Authors · 2022-11-16
> > > > **Thanks for the further comments.**
> > > >
> > > > - About "For the mask-based pretraining approach, first using it in MRL is not very impressive. The authors may remove it from the claimed contributions and organize the paper as a new Transformer-based backbone with the widely-used mask-based pretraining approach for 3D MRL."
> > > >   - Our paper didn't claim that we were the first masked-based 3D MRL approach.  What we claimed in the paper is "Based on extensive benchmarks, we build a simple and efficient SE(3)-equivariant Transformer backbone, and an effective 3D pretraining strategy in Uni-Mol."
> > > > - About "For experiments, what I really want to see is a clear table where the results are with the same training data and settings as the competitors and the data information is clearly included in the table. Last, only comparing Uni-Mol with GEM is not very convincing. More comparisons could be better. "
> > > >   - Most numbers in Table 1 and Table 2 are taken from the previous paper GEM, which was published at NMI. As GEM is directly compared to these numbers, and the setting of Uni-Mol is similar to GEM, we think our comparison to these numbers is also fair.
> > > >   - We cannot understand the comment "only comparing Uni-Mol with GEM is not very convincing". In Table 1 and Table2, there are several pretraining baselines besides GEM.
> > > > - About "The motivation of Transformer", and the visualization of self-attention
> > > >   - Our motivation of using Transformer is to capture the long-range interaction between atom pairs.
> > > >   - As shown in Fig. 7, there are several heads that learned the long-range interactions, like head 11 and head 13. In these heads, the attention weights for the atom pairs with large distances are large.
> > > >   - We believe the visualization explicitly reflects our motivation of using Transformer.
> > > > - About "Moreover, in the first submission, the visualization of self-attention is mainly about distance. However, in the revision, it becomes quite different, which makes me feel the paper is a bit unsolid."
> > > >   - In self-attention, there are multiple heads, and the patterns in different heads are usually different.
> > > >   - As we explained in the first response, we illustrated the average of all heads in our first submission. And the average pattern is very similar to the distance matrix.
> > > >   - In our revision, we further supplement the visualization by illustrating  the attention heads independently. And the average of all heads is also kept on the right-most side in Fig. 7.
> > > >   - In short, it is not "becomes quite different", but instead more results are supplemented.

---

> > > > > ### Comment · Reviewer_LVDL · 2022-11-17
> > > > > **Reply**
> > > > >
> > > > > Thanks for the follow-ups but they are not very clear. I still have a few questions.
> > > > >
> > > > > 1. Can the authors reclaim the technical contributions from the view of machine learning with detailed and specific novelties, such as the differences with AlphaFold-based methods?   There is a significant change between the initial submission and the revision.
> > > > >
> > > > > 2. Can the authors present a clear table to show all the information? It is difficult to associate the training dataset information (name and size) and the corresponding results from Table 1, Table 2 and the paper.

---

> > > > > > ### Author Response · Authors · 2022-11-18
> > > > > > **Further Response**
> > > > > >
> > > > > >
> > > > > > - About "There is a significant change between the initial submission and the revision. "
> > > > > >   - We believe we didn't change the technologies/methodologies in the revisions, and the revisions are highlighted by the color red. In particular, the revisions are mainly in the following two aspects.
> > > > > >     - More experiment results to demonstrate the effectiveness of the proposed method (mainly in the Appendix)
> > > > > >     - Clearer descriptions for the contributions/methods/figures/tables, avoiding misleading (mainly in page 2)
> > > > > >   -   Therefore, we don't quite understand what is "significant change", and hope the reviewer can specify it more clearly.
> > > > > > - About  "reclaim the technical contributions"
> > > > > >   - In all revisions, the contributions were listed in the last paragraph of the Introduction section.
> > > > > >     - 1) To our best knowledge, Uni-Mol is the first pure 3D molecular pretraining framework that can predict 3D positions, and the first molecular pretraining framework that can be directly used in 3D tasks in the field of drug design.
> > > > > >     - 2) Based on extensive benchmarks, we build a simple and efficient SE(3) Transformer backbone, and an effective 3D pretraining strategy in Uni-Mol.
> > > > > >     - 3) Uni-Mol outperforms SOTA in various downstream tasks.
> > > > > >     - 4) The whole Uni-Mol framework, including code, model, and data, will be made publicly available.
> > > > > >   - We don't quite understand the comment about "the differences with AlphaFold-based methods". Uni-Mol is an MRL framework, and we discussed the related works in the paper.  AlphaFold is a framework for protein structure prediction. They are designed for different applications. Maybe some components in AlphaFold's EvoFormer could be tried in future work, like Triangular Attention.
> > > > > > - About "a clear table to show all the information"
> > > > > >   - Thank you, we have made a table (refer to Table 24) as you suggest.

---

> > > > > > > ### Comment · Reviewer_LVDL · 2022-11-18
> > > > > > > **rebuttal reply**
> > > > > > >
> > > > > > > Thank you for Table 24. It is much easier for reference.
> > > > > > >
> > > > > > > The contributions you mentioned are mainly about applications. From the view of machine learning, alphafold integrates pair representation into self-attention and the proposed method has a similar design. Is there any significant difference?  Employing similar tools, designs or blocks to build the framework for a different application is not quite substantial. I would expect fundamental machine learning operations.
> > > > > > >
> > > > > > > Also, the main contribution is pretraining. However, the core of the method is based on a widely-used mask-based approach. Moreover, mask-based approaches are also used in 3D protein pretraining.  When you refer to existing works that claim to target pretraining or self-supervised learning in machine learning, you will see there is usually a novel pretraining or self-supervised learning approach, such as Jigsaw and MAE for image representation learning.
> > > > > > >
> > > > > > > I realize the accuracy improvement may have an impact on 3D MRL and I therefore will increase my score to 5. However, from the view of machine learning, I feel the novelty is limited and the score cannot be higher.  After all, ICLR is a machine-learning venue.

---

> > > > > > > > ### Author Response · Authors · 2022-11-19
> > > > > > > > **Thank you for the further reply!**
> > > > > > > >
> > > > > > > > We sincerely thank the reviewer for spending time discussing several rounds, very appreciated!
> > > > > > > >
> > > > > > > > - About the pair representation, and comparison with AlphaFold
> > > > > > > >   - Although with the same name ("pair representation"), they are actually quite different
> > > > > > > >     - In AlphaFold, the cost of maintaining pair representation is O(N^3), due to several triangular updates/attentions being used to update it.
> > > > > > > >     - In Uni-Mol, the cost of maintaining pair representation is negligible (Referring to Table 17), since there are no extra blocks/modules, only attention weight is used to update it.
> > > > > > > >     - So we believe it is not a similar design. Alphafold's pair representation is complex; Uni-Mol's pair representation is efficient, simple and effective (Referring to Table 14&15).
> > > > > > > > - About "mask-based approaches are also used in 3D protein pretraining. "
> > > > > > > >   - As we replied in the previous response (https://openreview.net/forum?id=6K2RM6wVqKu&noteId=2PARgwj2Nvn), most previous works in 3D MRL/protein pretraining focus on contrastive learning or distance/angle prediction. As far as we know, Uni-Mol is the first work to recover the 3D positions (coordinates) in pretraining.
> > > > > > > > - About "When you refer to existing works that claim to target pretraining or self-supervised learning in machine learning, you will see there is usually a novel pretraining or self-supervised learning approach, such as Jigsaw and MAE for image representation learning. "
> > > > > > > >   - At the concept level, mask approaches are to recover the original data based on the corrupted inputs. From this point of view, MAE and Jigsaw (the papers you mentioned) are also mask approaches: The task of MAE is to recover the masked pitches; the task of Jigsaw is to predict the correct permutation of pitches based on random-permuted inputs.
> > > > > > > >   - In Uni-Mol, the task is to recover atoms' types and coordinates, given the corrupted atom types and coordinates (we consider the possible permutations due to the noise in coordinates).
> > > > > > > >   - Therefore, we are quite confused about the comment. Both Uni-Mol, MAE, Jigsaw are "widely-used masked approaches", why are MAE and Jigsaw novel, but Uni-Mol is not?
> > > > > > > > - About novelty.
> > > > > > > >   - We think it is hard to define and evaluate  "novelty", as different people have different perspectives. Therefore, in our paper, rather than the subjective evaluation "novel", we mostly use objective evaluations (like "simple", "efficient", "effective"). And extensive experiments are designed to demonstrate them.
> > > > > > > >   - Therefore, we understand and appreciate the reviewer's comments about the novelty. And we also very much appreciate that the reviewer increased the score due to Uni-Mol's improvement in performance.
> > > > > > > >
> > > > > > > > We thank the reviewer again for the valuable comments and suggestions.

---

> ### Comment · Reviewer_CPRe · 2022-11-17
> **Updated review**
>
> 1) I realized, thanks to reviewer LVDL, that the novelty was questionable, as pointed in his weakness point (1). As Authors answer, then, Uni-Mol is new in its ability to deal with 3D downstream tasks, but it's important that the claims were reduced.
>
>    Because of this, I still support the paper but decrease a bit my grading, from "should be highlighted" to "good paper, should be accepted".
>
>
> 2. Furthermore, I agree with referee LVDL that the claim about the SE(3)-equivariant transformer is a bit over-claimed, or weak/misleading.
> More precisely, I would complain again about the vocabulary, equivariance vs invariance (sorry for my lapsus in my previous review).
> Here it's true that Eq. (3) is SE(3)-equivariant wrt positions x, but in a trivial way, because you just multiply (equivariant) input (x_i-x_j) with an invariant number c_ij.
>
>     It is not like the architecture of the backbone is designed in a subtle way to propagate equivariant features, as is possible using approaches as those described in [Tensor field networks: Rotation- and translation-equivariant neural networks for 3D point clouds. arXiv:1802.08219] , or the library [https://e3nn.org/].
>
>      So, I would revise the wording, to avoid misleading/confusing readers.
>     I would replace, everywhere:
>     "SE(3)-equivariant Transformer [backbone/architecture]" ->  "SE(3)-invariant Transformer [backbone/architecture]"
>     And when introducing Eq.(3), I would add a few words to be more explicit about the origin of equivariance, something like:
>     > To this end, we introduce an SE(3)-equivariance head
>
>     [that combines the equivariant input (xi-xj) with the SE(3)-invariant (pair-wise?) attention coefficient]
>
>     > to predict the delta positions based on pair representation,
>
>      (btw you write "SE(3)-equivariance head" but should write "SE(3)-equivariant head", I think)
>
>      Yet, the fact that "it works" shows that the work is good and should be acknowledged, but the use of words is, in my opinion, misleading, because one could imagine using more deeply-equivariant architectures, i.e. archis which propagate equivariant features through layers (not what is done here).
>
>
> 3. I agree with LVDL that the fact it's a mask-based approach is not much new (although for 3D molecules, going from masking 2D to masking 3D is not a conceptual novelty).
> I liked the way you introduced it, comparing with NLP etc, but it should not be oversold.
>
>
> 4. I like the visualization of attention heads and table 18 (which however lacks the bold font).
> It could be interesting to comment quickly on the fact that it can happen to have fully-connected (transformer) that performs worse than local-connected graph, as in BACE.
>
>
> 5. About pre training or lack thereof:
> - I think it's worth repeating in table's caption, when Uni-mol is compared to baselines without pre-training. People skim through papers, reading only figures and tables, so self-sufficient captions are important.
> - I am a bit confused about Uni-Mol_random. Is it that the backbone is not trained at all, and used as initialized for learning downstream tasks (with frozen weights in the backbone), or is it that it's trained, but only on the downstream task (starting from random initialization). I would expect it's the second one, but I am not sure, which shows it should most likely be explicited.
>
>
> Thank you for your detailed answers and improved revision.

---

> > ### Author Response · Authors · 2022-11-17
> > **Thank you so much for the further comments!**
> >
> > Thank you very much for the further comments and the support of our paper! We have further revised the paper.
> >
> > - About the wording of "SE(3)-equivariant transformer".
> >   - We are sorry, we didn't realize it would cause the misleading. We thought  the architecture could predict SE(3)-equivariant positions could be called "SE(3)-equivariant [backbone/architecture]".  We will revise the wording.
> >   - We also want to highlight, despite its simplicity,  the proposed backbone is very efficient, and effective in conformation generation: 1)  Table 17 shows our proposed backbone is much faster than SE(3) Transformer[1] ; 2) Table 21 shows  our proposed backbone (without pretraining & with the same model capacity) is much better than SE(3) Transformer[1] in conformation generation.
> > - About the result of BACE.
> >   - The result indeed is very interesting: Although the local-connected graph is not good in most tasks, it outperforms the full-connected graph (Transformer) in BACE.
> >   - The BACE dataset provides qualitative (binary label) binding results for a set of inhibitors of human β-secretase 1. That is, the BACE task is to predict the binding affinities of a set of molecules, given a protein. From the view of chemistry, the binding affinities mainly depend on the interactions between a group of atoms, known as functional groups, on the ligand and the residues on the protein. In other words, a part of the molecule, not the whole molecule, is critical in the BACE task. Therefore, with the local-connected graph, the model can focus more on local atoms, rather than all atoms, and then may perform better than the full-connected graph in the BACE task.
> > - About the table caption for non-pretraining baselines.
> >   - We have revised the captions of tables that contain non-pretrained methods.
> >   - "uni-mol_{random}" is trained by downstream data started from random initialization.  We have updated all "uni-mol_{random}" to "uni-mol_{no_pretrained}" for better clarity.
> >
> > [1] Fuchs, Fabian, et al. "Se (3)-transformers: 3d roto-translation equivariant attention networks." Advances in Neural Information Processing Systems 33 (2020): 1970-1981.

---

### Official Review · Reviewer_uSFd · 2022-10-24

**Confidence:** 4
**Clarity, Quality, Novelty And Reproducibility:** The quality, clarity, and originality…
**Correctness:** 4
**Technical Novelty And Significance:** 3
**Empirical Novelty And Significance:** Not applicable
**Recommendation:** 8

**Strength And Weaknesses:**

##########################################################################

Pros:

- Uni-Mol is the first pure 3D molecular pretraining framework, and the first molecular pretraining framework that can be directly used in 3D tasks in the field of drug design. which is a very valuable work for 3D molecular representation learning.
- Uni-Mol contains a simple and efficient SE(3)-equivariant Transformer backbone, and an effective 3D pretraining strategy. The ablation benchmarks demonstrate their superior performance, and Uni-Mol also outperforms SOTA in various downstream tasks.
- The paper is well-written and the experiment section is solid.
##########################################################################

Cons:

- I think this paper is a solid work, a small piece of advice is since this paper focuses on 3D molecular data, in addition to quantitative experimental results, it will be more intuitive and convincing if more qualitative results can be displayed. Another concern is how much computational resources and time for 3D molecular pretraining needs to consume, which the paper does not seem to mention.

**Summary Of The Paper:**

This paper proposes a universal 3D Molecular representation learning (MRL) framework, called Uni-Mol, that significantly enlarges the representation ability and application scope of MRL schemes. Uni-Mol contains two pretrained models with the same SE(3)-equivariant transformer architecture: a molecular model pretrained by 209M molecular conformations; a pocket model pretrained by 3M candidate protein pocket data. Besides, Uni-Mol contains several finetuning strategies to apply the pretrained models to various downstream tasks. By properly incorporating 3D information, Uni-Mol outperforms SOTA in 14/15 molecular property prediction tasks. Moreover, Uni-Mol achieves superior performance in 3D spatial tasks, including protein-ligand binding pose prediction, molecular conformation generation, etc.

**Summary Of The Review:**

good paper.

---

> ### Author Response · Authors · 2022-11-10
> **Response to Reviewer uSFd**
>
> Thank you so much for supporting our work and valuable review comments! We have revised the paper according to your suggestions.
>
> > since this paper focuses on 3D molecular data, in addition to quantitative experimental results, it will be more intuitive and convincing if more qualitative results can be displayed
>
> Besides the visualization of protein-ligand complex prediction in Appendix F, we add more visualization results:
> 1.  More attention maps (Appendix H), demonstrating the different learned interaction patterns among atom pairs;
> 2.  Visualization of the 3D position recovery task (Appendix I), demonstrating the effectiveness of the proposed pretraining tasks.
>
> > how much computational resources and time for 3D molecular pretraining needs to consume, which the paper does not seem to mention.
>
> Molecular pretraining costs about 20 hours on 8 V100 GPUs. Pocket pretraining costs about 2 days and 20 hours by 8 V100 GPUs. We have updated these numbers to Appendix C.1 and C.2.
>
> We appreciate the reviewer for spending time reviewing our paper and offering valuable suggestions. If you have any further questions, please tell us and we are willing to address your concerns.

---

### Official Review · Reviewer_CPRe · 2022-10-25

**Confidence:** 4
**Correctness:** 3
**Technical Novelty And Significance:** 3
**Empirical Novelty And Significance:** 4
**Recommendation:** 10

**Clarity, Quality, Novelty And Reproducibility:**

Clarity:
- overall, paper is clear.
- the architecture explanation could be more detailed. For instance the node representation initialization is not specified (I guess it includes atom type, and not positions, for equivariance?)

Quality:
- overall, the statements are sound. A few remarks:
- I think the embedding of edges (atom pairs) into a (type-aware) Gaussian Kernel function of the norm of the distance of atom-pair, is an invariant representation, not an invariant one. Then, Eq (3) may indeed be equivariant , because c_ij is invariant and x_i is of course equivariant. I think the mention of equivariance should be corrected, as it is confusing for the reader.
- atom types are a bit hidden. In Fig 2, instead of atom head, it could read "type head" (I suggest..)
- I am from GNNs. I am not very familiar with transformers, I just know the attention mechanism (esp. the standard softmax one used here).
    It seems to me like your transformer can be seen as an attention-GNN with no aggregation mechanism for the edge features (just a ResNet style addition) and a new node value that solely depends on the neighbor edges (via attention, Q-K-V). In interpret the Value as the node feature.  If I am not too much wrong, then I think it would be a good correction to show how the architecture presented fits into the GNN framework. It seems to me that it does not differ much from it. (also, you use a fully-connected graph).

    The current introduction to the paper is otherwise slightly misleading:
    > Most previous MRL frameworks used graph neural networks(GNN) [22; 23; 12]  (...) Therefore, we use Transformer as the backbone
model to fully connects nodes/atoms.
- runtimes of pre-training are missing (or I missed them, not highlighted enough)


Novelty:
- as stated in my summary of the paper, there are several original contributions, but no new theoretical one

Reproducibility:
- for now, the code is not publicly available. It is stated it will be. Given the paper's goal, I believe it will !


**Strength And Weaknesses:**

Strengths:
- the contribution is both original (new) and strong.
- performance is extensively review (many downstream tasks, enough detail to follow)
- the simplicity of the ideas: mostly using and combining already existing conceptual tools, in general rather clearly explained (from someone with a good ML/DL background)

Weaknesses:
- no major conceptual novelty (may be an asset in a sense)
- Architecture is a bit overlooked/not explained enough in the main text (page 3)
- a few unclear parts, or parts that need a bit more discussion


**Summary Of The Paper:**

This paper proposes to push representation learning for molecule-related task further than before, by mixing different tasks in its per-training step, defining a representation that is generic enough to be applicable to a variety of downstream tasks, on which it performs as well or better (and several times, significantly better) than the reported SOTA.
The main contributions are :
- this original idea of mixing very different pre-training tasks / going further in representation learning
- the extensive per-training performed, in terms of the strategy: the model used is the transformer, and the strategy is to imitate the guess-a-missing-word strategy: instead, one needs to recover the corrupted position of each atom.
- the extensive per-training performed, in terms of training data (in my perspective it seems impressive, but I haven't read the related literature)

What is not new, as paper admits, is:
- The transformer architecture itself
- the various type, position, relative distances encoding (although they are well benchmarked), that are SE(3)-invariant
- the tasks and datasets (taken individually), except for a regression task on pocket property prediction, which is new (no Baseline there).


**Summary Of The Review:**

Aside from a few points that need further correction/clarification/discussion, the paper is sound and good, and brings a new way of pre-training 3D representations to the community, with proved efficiency.

---

> ### Author Response · Authors · 2022-11-10
> **Response to Reviewer CPRe**
>
> Thank you so much for supporting our work and valuable review comments! We have revised the paper according to your suggestions. We (temporally) highlight the updates with the red color in the paper, for your convenience to check the changes. In particular, we made the following changes.
> - We revised the first paragraph in Sec 2.1, to more clearly describe why we choose Transformer.
> - We add an "Architecture Overview" paragraph in Sec 2.1, to explain the model's inputs and how atom/pair representation is initialized.
> - We also updated Figure 2, changing the "atom head" to "atom type head".
> -  We change the "spatial positional encoding" to "invariant spatial positional encoding", to highlight its invariance.
> - We add the computational cost of the pretraining to Appendix C.1 and C.2.
>
> If there are still unclear parts, please let us know.
>
> Following are our responses to the questions.
>
> > node representation initialization is not specified
>
> You are right. The node/atom representation is initialized by atom types, with Embedding layers. We also revised the paper to make it clearer.
>
>
> > I think the embedding of edges (atom pairs) into a (type-aware) Gaussian Kernel function of the norm of the distance of atom-pair, is an invariant representation, not an invariant one.
>
> Maybe it says "is an invariant representation, not an equivariant one."
> Yes, you are right. The pair representation is invariant, not equivariant. We have revised the paper, changing the "spatial positional encoding" to "invariant spatial positional encoding", to emphasize the invariant of the position encoding.
>
> > It seems to me like your transformer can be seen as an attention-GNN with no aggregation mechanism ... It seems to me that it does not differ much from it. (also, you use a fully-connected graph).
>
> We have revised the paper by changing the first paragraph in Sec. 2.1, to avoid possible misleading. We choose Transformer mainly because it can handle the fully connected graph naturally, and thus can learn the long-range interaction. And you are right. When with fully connected graphs, GNN does not differ much from Transformer.  And we would like to extend Uni-Mol to the GNN framework in future work.

---

### Official Review · Reviewer_6Ybw · 2022-11-14

**Confidence:** 4
**Correctness:** 4
**Technical Novelty And Significance:** 3
**Empirical Novelty And Significance:** 3
**Recommendation:** 8

**Clarity, Quality, Novelty And Reproducibility:**

As pointed earlier, it is a good read and the model is well engineered. The novel SE(3) equivariant that is efficient and simple to implement is central to the SOTA results. Given that whole lot of details are being provided in the appendix it should be possible to reproduce the results.

**Strength And Weaknesses:**

Strengths:
1. The paper is well-written and easy to follow. As per my knowledge, it is one of the first work to have shown to incorporate 3D information to improve representation capacity of molecular processing architecture.
2. In order to preserve rotation and translation invariant properties, Uni-Mol totally operates only on input atomic representation and pairwise Euclidean distance inputs.
3. Unlike the previously introduced SE(3) equivariant transformers that uses complex processing blocks, Uni-Mol attention layer is quite simple that transforms to and fro between pairwise and atomic representations.This is possible by relegating the 3D positions update to the final layer of the network.
4. With efficient backbone, Uni-Mol is able to effectively pretrain on large-scale molecular and protein pockets datasets.

Weaknesses:
When drawing evaluation against various baselines on benchmark tasks, it is quite unfair to make comparison of models trained solely on task-specific data to Uni-Mol which has been pretrained on large datasets. For instance,
a. On molecular conformation tasks, all other baselines generates conformation using only 2D molecular graph information. On the other hand, Uni-Mol leverages the 3D position output from RDKit (that uses ETKGD with MMFF) as input for pairwise representation. This for sure provides better start-point for Uni-Mol compared to other baseline that starts with gaussian random noise as 3D position.
b. Moreover, pretrained Uni-Mol architecture is relatively very high capacity when compared to baselines.

Given the nature of contribution it may be difficult to establish apple to apple comparisons. However, fair comparison the efforts can be put into establishing,
a. similar capacity Uni-Mol baseline network
b. Uni-Mol baseline that is evaluated using gaussian noise input
c. evaluating each task using other MRL networks


**Summary Of The Paper:**

This work proposes a novel universal molecular representation learning framework, called Uni-Mol, that incorporates 3D information. The network consisting of stack of SE(3)-equivariant transformer layers, is pretrained on two different tasks each involving molecular and protein pocket data respectively. This pretraining is established by learning to predict 3D atomic positions and masked atom representation from the noisy input. These pretrained networks when applied for finetuning various downstream tasks, it is shown to achieve superior performance on molecular / pocket property prediction, molecular conformation generation and protein-ligand binding pose prediction.

**Summary Of The Review:**

Overall, it is a good work and I am inclined towards accepting this work. It establishes two different models that can be put to use for improving accuracy in many downstream tasks. It also provides good tricks for setting up pretraining tasks on molecular / drugs data. For now, I see Uni-Mol strengths outbeats its weaknesses and its contribution may open up avenue of progress in drugs discovery.

---

> ### Author Response · Authors · 2022-11-15
> **Response to Reviewer 6Ybw**
>
> Thank you so much for supporting our work and valuable review comments!
>
> We agree that the experiment in the molecular conformation generation is somehow not fair, and we further set up an experiment, shown in Appendix E.9.
> In particular, we add a baseline, Guan et al.[1], which used a SE(3) Transformer to optimize the RDKit-generated conformation to the DFT conformation (GEOM dataset). This setup is the same as Uni-Mol in molecular conformation generation. To have a fair comparison, we also set up a small Uni-Mol model (the same capacity as Guan et al.) without pretraining. The result is shown in Table 21. From the result, we can see that Uni-Mol outperforms Guan et al. significantly in a fair comparison. The result indicates the effectiveness of the backbone model in Uni-Mol. In addition to this, we find Uni-Mol with more parameters is slightly worse than the small one. We think it is due to the over-fitting of the large model without pretraining.
>
> We sincerely hope that the additional experiment addresses your concerns. We are also willing to discuss with you if you have any further concerns.
>
> Reference:
> [1] Jiaqi Guan, Wesley Wei Qian, Wei-Ying Ma, Jianzhu Ma, Jian Peng, et al. Energy-inspired molecular conformation optimization. In International Conference on Learning Representations, 2022

---

### Author Response · Authors · 2022-11-10
**Thanks all reviewers!**

We appreciate the reviewers for spending time reviewing our paper and offering valuable suggestions, and we have revised the paper according to the comments. We (temporally) highlight the updates with the red color in the paper, for your convenience to check the changes. If there are still unclear parts, please let us know.

There is one thing we want to clarify here, that is Reviewer LVDL said in Weakness 2, the experiment comparison is not fair, since Uni-Mol uses more data (209M conformations). We believe the comparison is fair, details are in the following:
- As described in Sec. 2.2, the number of molecules used in Uni-Mol is **19M**, the same scale as GEM (20M). Based on the 19M molecules, we generate 11 conformations for each molecule, in total 209M conformations. As a molecule has several different conformations naturally, using multiple conformations is a reasonable data augmentation in 3D MRL.
- GEM also uses 10 conformations for each molecule, **in total 200M conformations**. refer to their code:  https://github.com/PaddlePaddle/PaddleHelix/blob/ee1acdb6b3bb6d6cb9cf8c27373669e5c79605c0/pahelix/utils/compound_tools.py#L674 and https://github.com/PaddlePaddle/PaddleHelix/blob/ee1acdb6b3bb6d6cb9cf8c27373669e5c79605c0/pahelix/featurizers/gem_featurizer.py#L213
- Combining the above, the comparison with GEM (200M conformations) is fair.
- We also conduct an additional experiment that only uses 1 conformation for each molecule, referring to Table 18 in the paper. The result is only slightly worse than the one with 11 conformations, and still outperforms GEM in most tasks.

---

### Decision · Program_Chairs · 2023-01-20

**Decision:**

Accept: poster

**Justification For Why Not Higher Score:**


This is a simple and effective SE(3) equivariant transformer architecture. The mechanism ensuring equivariance is quite simple / lightweight. The tasks themselves are not particularly novel but brought together coherently.

**Justification For Why Not Lower Score:**


The evaluations are quite comprehensive, covering molecular property prediction, conformer generation, pocket property prediction, as well as protein-ligand binding pose prediction tasks. Consistent gains from pre-training shown across tasks.

**Metareview: Summary, Strengths And Weaknesses:**


The paper proposes a simple and effective SE(3) equivariant transformer architecture for pre-training molecular representations. Emphasis is on the efficiency of the "backbone model" and this allows one to pre-train it on larger 3D datasets. The mechanism ensuring equivariance is quite simple. The architecture is trained on both small molecule conformers as well as (separately) on protein structures/pockets. The pre-training tasks include coordinate recovery from corrupted inputs, pair-distance prediction, and atom type recovery from masked entries. The tasks themselves are not particularly novel but brought together coherently. The evaluations are quite comprehensive, covering molecular property prediction, conformer generation, pocket property prediction, as well as protein-ligand binding pose prediction tasks. During fine-tuning for protein-ligand pose prediction, the pocket and ligand representations are first obtained independently from their respective pre-trained models, then combined as an input to a pair-distance predictor for scoring. While the interaction model could be improved, the setup already enables direct back-propagation for pose coordinate adjustment during testing. As discussed in the reviews, conformer generation with pre-training has a clear advantage to baselines trained only on task data. Additional analysis provided by the authors do suggest an advantage even absent pre-training and with matched capacity.


**Note From Pc:**

if the above contains the word "oral" or "spotlight" please see: "oral" presentation means -> notable-top-5% and "spotlight" means -> notable-top-25%. As stated in our emails, we are disassociating presentation type from AC recommendations

**Summary Of Ac-Reviewer Meeting:**